

# Reduction of airmass-dependent biases in TCCON XCH$_4$ retrievals during polar vortex conditions

Jonas Hachmeister[1], Debra Wunch[2], Erin McGee[2], Kimberly Strong[2], Rigel Kivi[3], Justus Notholt[1], Thorsten Warneke[1], and Matthias Buschmann[1]

[1]Institute of Environmental Physics (IUP), University of Bremen, Bremen, Germany
[2]Department of Physics, University of Toronto, Toronto, Canada
[3]Finnish Meteorological Institute, Helsinki, Finland

**Correspondence:** Jonas Hachmeister (jonas_h@iup.physik.uni-bremen.de)

**Abstract.** Trace gas measurements from the Total Carbon Column Observing Network (TCCON) are important for monitoring the global climate system and for validating satellite measurements. In the Arctic, ground-based data coverage is relatively limited due the inherent challenges of conducting measurements in this region (e.g., remoteness, harsh weather) and the polar nights, which prevent solar absorption measurements for half of the year. TCCON measurements from the Arctic sites are of

significant value for the validation of satellite data products in this region, as these measurements can extend the spatio-temporal coverage in the Arctic. In this study, we investigate the TCCON methane (CH$_4$) retrieval under polar vortex conditions. The CH$_4$ profile exhibits a distinct shape inside the vortex, which is related to the descent of stratospheric air inside the vortex. We show that the standard TCCON CH$_4$ prior does not sufficiently reproduce this profile shape, leading to airmass dependencies (AMDs), increased spectral residuals and less sensitive averaging kernels. These effects can be explained by the fact that

TCCON uses a profile scaling retrieval (PSR) where the prior shape is fixed and only a scaling factor is retrieved. We further show that changes in the prior can improve the retrieval within the polar vortex. This leads to mean differences between 1 and 2 ppb in XCH$_4$ compared to the standard retrieval, and maximum differences up to roughly 17 ppb. This manuscript highlights the importance of understanding the limitations of retrieval methods to avoid misinterpretation of data. Furthermore, it emphasizes the need to investigate the shape of trace gas profiles inside the polar vortex to improve PSR in the Arctic, which

could include in situ data campaigns focusing on inside-vortex air.

## 1 Introduction

The Total Carbon Column Observation Network (TCCON) is a global network of Fourier Transform Spectrometers (FTSs) that measure various atmospheric constituents using direct solar spectra in the near-infrared spectral region. TCCON measurements adhere to a high quality standard and provide valuable information about the chemical composition of the atmosphere. The

column-averaged dry-air mole fractions of atmospheric gases are derived using a profile scaling retrieval which works well for atmospheric gases for which the profile shape is sufficiently well known. It is thus of great importance to continuously monitor and improve the used prior shapes to ensure a good working retrieval. Currently, the TCCON network includes three stations north of the Arctic Circle: Sodankylä (67 °N), Ny-Ålesund (79 °N) and Eureka (80 °N). A fourth station, Cambridge Bay (69





°N), is to be added to the network. To analyze incursions of polar vortex filaments, which are especially present during the
breakdown of the vortex in spring, we include East Trout Lake (54 °N) in our analysis. The stratospheric polar vortex forms
during late autumn and breaks down in early spring. Air inside the polar vortex is relatively well isolated from mid-latitude
air and exhibits a distinct chemical composition, due to the descent of upper stratospheric air inside the vortex. This leads to
significantly different vertical profile shapes for gases that have different chemistry between the troposphere and stratosphere,
such as methane ($CH_4$). Due to the decreasing $CH_4$ concentration with height above the tropopause, the descent of air leads to
a reduction of $CH_4$ at lower altitudes, which causes a significantly different profile shape compared to out-of-vortex conditions
(see e.g., Fig. 6).

Issues with inside-vortex TCCON retrievals have been previously reported, e.g., for $N_2O$ by (Zhou et al., 2019; Vanden-
bussche et al., 2022) and for $CH_4$ (Ostler et al., 2014; Tukiainen et al., 2016). In this manuscript, we systematically investigate
the impact of the polar vortex on the TCCON $XCH_4$ retrievals and formulate steps to improve the retrieval by modifying the
$CH_4$ prior profiles. Section 2 provides a short overview of the TCCON network and data. Section 3 introduces the polar vortex
and explains how it can be located. In Sec. 4, we examine the airmass dependence (AMD) present in Arctic TCCON $XCH_4$
data and provide an explanation for the enhanced AMD when measuring polar vortex air. Section 5 briefly looks at existing
measurements of inside-vortex $CH_4$ profiles and compares these to the TCCON $CH_4$ prior. In Sec. 6, we lay out our methods
for modifying the $CH_4$ prior. Finally, we present our results in Sec. 7 and our Conclusions in Sec. 8.

## 2  TCCON $XCH_4$ data

The TCCON data are retrieved from direct solar spectra measured by a FTS in the near-infrared spectral region. The dry-air
column-averaged mole mixing rations of atmospheric gases are derived using a profile scaling retrieval (Wunch et al., 2011).
The retrieval algorithm compares the measured spectra to spectra calculated using the forward model and the (scaled) prior
profile. During this iterative process, the prior profile is scaled to minimize the cost function. The algorithm thus only retrieves
the total column and no profile information. This approach is significantly faster and more stable than a profile retrieval (Roche
et al., 2021) and works well for atmospheric gases for which the profile shape is sufficiently well known. The measurement
precision error for the column-averaged dry-air mole fraction of methane ($XCH_4$) is reported to be generally below $0.3\%$
(roughly 5 ppb, Laughner et al. (2024)). TCCON measurements are adjusted using an airmass-dependent correction factor,
derived offline from the data itself and an airmass-independent correction factor (AICF). The AICF is determined by com-
parisons with in situ profiles measured over TCCON sites by aircraft or balloon payloads. The AICF for $XCH_4$ is tied to the
World Meteorological Organization (WMO) X2004 calibration scale. Satellite data are in turn often validated using TCCON
measurements and thus indirectly tied to the WMO scale (Wunch et al., 2010; Laughner et al., 2024).

In this manuscript, we use data from the stations in East Trout Lake (ETL, 54 °N, Wunch et al. (2011)), Sodankylä (SOD,
67°N, Kivi and Heikkinen (2016); Kivi et al. (2022)), Ny-Ålesund (NYA, 79 °N, Buschmann et al. (2022)) and Eureka (EUR,
80 °N, Strong et al. (2022)).





## 3 The stratospheric polar vortex

The stratospheric polar vortex is a band of westerly winds which begins to form around the poles after the fall equinox. The cooling of the stratosphere, caused by the absence of sunlight, leads to the descent of air and hence a pressure gradient compared to the mid-latitudes. Combined with the rotation of the earth (Coriolis effect), this leads to the formation of westerly winds around the poles (Schoeberl and Hartmann, 1991; North et al., 2014). The zonal East-West flow usually becomes well organized by mid-November (North et al., 2014). The stratospheric polar vortex is not to be confused with the tropospheric circumpolar vortex (polar jet stream) at 500hPa that is present for the whole year (Serreze and Barry, 2014).

Air enters the stratosphere in the tropics and is transported up and toward the pole through the so-called extra tropical pump (Holton et al., 1995). The air in the upper stratosphere is well isolated from the troposphere and attains a distinct chemical composition. Descent of air during the Arctic winter transports the chemical composition of the upper stratosphere to the lower stratosphere, while lower stratospheric air masses are displaced towards the mid-latitudes. Since the lower stratosphere can exchange air with the troposphere by transport along isentropic surfaces (Holton et al., 1995; Brasseur et al., 1999) its composition significantly differs from the upper stratospheric air.

While the air outside the vortex is laterally mixed through planetary waves, mostly negating the effect of the descent, the air inside the polar vortex is well isolated (Schoeberl and Hartmann, 1991; North et al., 2014). This leads to a chemically distinct composition of stratospheric vortex air compared to mid-latitude air masses (Schoeberl and Hartmann, 1991; Nash et al., 1996; Ostler et al., 2014; North et al., 2014; Karppinen et al., 2020). This air can also leave the polar vortex in the form of filaments, as planetary waves erode the vortex edge (Waugh et al., 1994; Whaley et al., 2013; North et al., 2014) or when the polar vortex breaks up in early spring.

In contrast to the relatively stable Southern Hemisphere polar vortex, its northern counterpart exhibits more variability in strength, covered area and duration. This difference is due to the higher planetary wave activity in the Northern Hemisphere, caused by the more varied orography (Schoeberl and Hartmann, 1991; North et al., 2014).

Trace gas measurements using remote sensing techniques based on solar absorption spectroscopy (like TCCON or various satellites) are expected to be affected by the polar vortex only in (early) spring, when sufficient light again becomes available to conduct measurements, as the vortex needs time to fully form during the autumn.

### 3.1 Locating the polar vortex

#### 3.1.1 Polar vortex mask based on Nash criterion

The evolution of the polar vortex can be monitored using potential vorticity (PV), which is defined as the dot product of the absolute vorticity and the gradient of some conservative thermodynamic property (e.g., potential temperature). For adiabatic or isentropic moving air parcels this quantity is conserved, making the analysis of PV on potential temperature surfaces a potent tool in analyzing the structure of the polar vortex (Schoeberl and Hartmann, 1991; Nash et al., 1996).

Here we use the criteria formulated by Nash et al. (1996) to calculate the polar vortex boundary from ERA5 data (Hersbach et al., 2017). Specifically, we use potential vorticity and u-wind data on potential temperature levels with a $0.25° \times 0.25°$





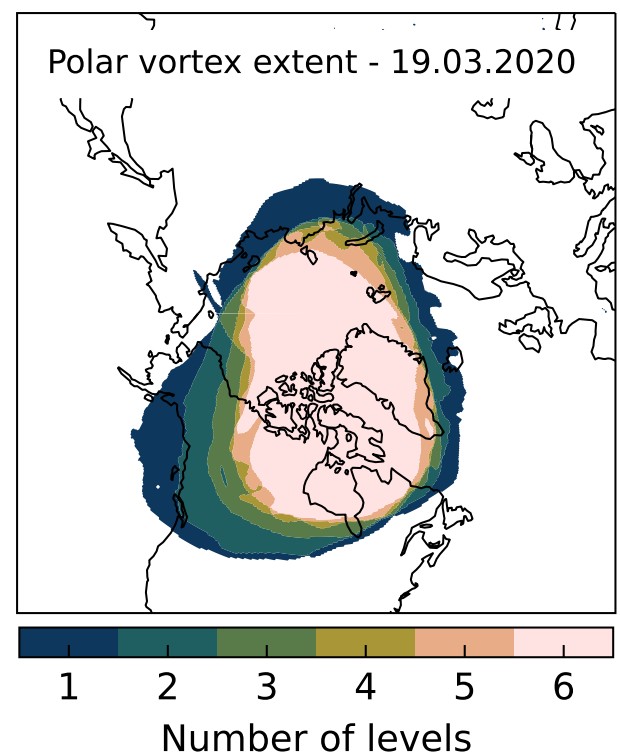

**Figure 1.** Combined polar vortex mask for 19 March 2020. The contour values correspond to the number of potential temperature levels on which the polar vortex is detected. The polar vortex masks are caluclated on six potential temperature levels between 395 K and 700 K.

resolution. Since the polar vortex is a three-dimensional structure, we calculate masks on six potential temperature layers between 395 K and 700 K, and use the combination of all masks to determine whether a measurement is "inside" the polar vortex. See Appendix A for a detailed explanation. Figure 1 shows an example of the polar vortex mask for a single day. Measurements that are outside all masks are considered as "out-of-vortex". Measurements that are inside the vortex boundary on three or more masks are considered "inside-vortex". This threshold was chosen empirically as it provides a robust mask while not constraining the vortex extent too much. Note that the polar vortex has a vertical structure, hence there is no definite criterion on what qualifies as inside or outside the vortex.

### 3.1.2 Detection of polar vortex air using a chemical tracer

Another method for identifying whether a certain location is inside the polar vortex, is by using a chemical tracer with known behavior inside the polar vortex. Such a gas is hydrogen fluoride (HF), which increases with height in the stratosphere, making it an ideal tracer for atmospheric dynamics in this region (Chipperfield et al., 1997). HF is formed in the stratosphere through reactions of fluorine with methane, hydrogen or water vapor. The main source of fluorine is the photolysis of CFCs (Brasseur





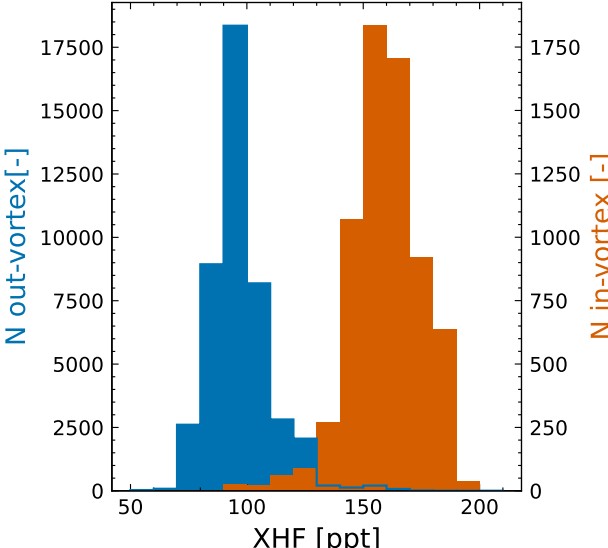

**Figure 2.** Histograms of in- and out-of-vortex XHF from TCCON Ny-Ålesund data. The histograms for in- and out-of-vortex data are well separated, indicating that XHF can be used as a proxy for the presence of vortex air. The in- and out-of-vortex assignment was done using a polar vortex mask generated from ERA5 data.

et al., 1999). HF is very stable in the stratosphere and its distribution is largely determined by the distribution of its sources and atmospheric dynamics. In the tropical stratosphere mixing ratios of HF are generally smaller for a given height due to the strong upward transport of tropospheric air, while concentrations in the high latitudes are generally larger due to the down-welling of air. This effect is particularly strong inside the polar vortex (Brasseur and Solomon, 1984). HF is removed from

105 the troposphere by dry and wet deposition (Cheng, 2018). As HF is effectively removed in the troposphere by wet deposition, mixing ratios increase with height in the stratosphere. The descent of air inside the polar vortex can be identified with increases in the total column of HF (XHF). XHF is measured by TCCON and thus enables the detection of vortex air. While it doesn't provide information on the full 3D extent of the polar vortex, it allows for the detection of filaments, i.e. air masses separated from the polar vortex, that are often not captured in the vortex masks.

Figure 2 shows histograms of TCCON NYA XHF that have been categorized as in- or out-of-vortex using the vortex mask described above. The histograms are clearly separated and indicate that XHF can be used as a tracer for polar vortex air. The small number of high XHF out-of-vortex measurements are probably measurements of air masses separated from the vortex, which are not visible in the vortex mask. The low XHF inside-vortex measurements are likely related to difference in descent between the boundary and center of the vortex and the variability in descent between different years. Additionally, the quality

of the vortex mask is limited by the quality and resolution of the ERA5 data, which can explain some of the potential false positives or negatives.





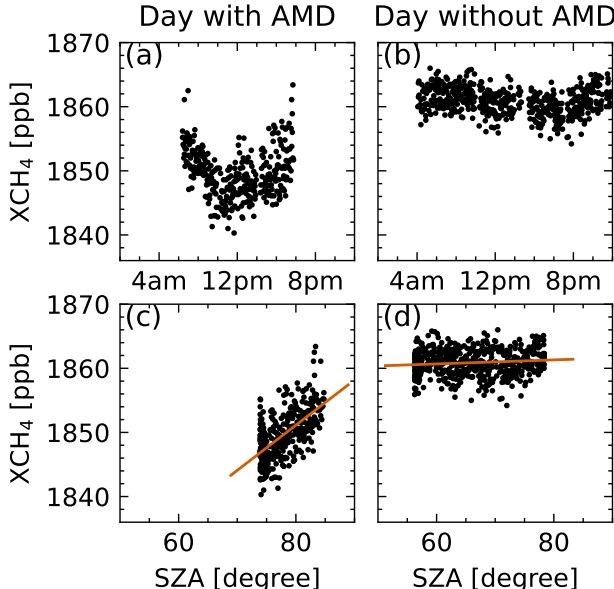

**Figure 3.** Panel (a) and (c) showcase XCH$_4$ for a day with AMD. Panel (b) and (d) showcase XCH$_4$ for a day without AMD. Data are from the Ny-Ålesund TCCON station. Panel (a) shows a day with AMD, as visible in the u-shaped XCH$_4$ centered around the local noon. The AMD is also visible in the linear relationship between XCH$_4$ and SZA in panel (c). Panel (b) shows a day without AMD and hence no visible symmetry around noon. Consequently there is no clear slope in panel (d).

## 4 Airmass dependence of TCCON XCH$_4$

All TCCON data have a solar zenith angle (SZA) or airmass-dependent artifact that causes the retrieved column to be smaller or larger for high SZA compared to low SZA measurements. The airmass dependence (AMD) is caused by airmass-dependent variation in the retrieved target gas and/or O$_2$ in such a way that both effects don't compensate. AMDs can be caused by uncertainties in spectroscopy, by instrument alignment, by non-linearity problems and by the use of the wrong measurement time. TCCON data are corrected during post-processing using an airmass-dependent correction factor that is derived empirically and applied consistently throughout the network; that is, a single airmass-dependent correction is applied to each gas for every instrument in TCCON. See Wunch et al. (2011) and Laughner et al. (2024) for more information.

XCH$_4$ from Arctic TCCON stations can sometimes exhibit a significant residual airmass dependence that is not corrected by the network-wide airmass-dependent correction. (Inverted) U-shapes of diurnal XCH$_4$, which are symmetric around noon are a strong indicator for an AMD, since there is no reasonable geophysical explanation for diurnal XCH$_4$ variations of this magnitude. Diurnal variability in the Arctic winter and spring is relatively weak and most likely driven by transport from lower latitudes (AMAP, 2015). This cannot account for the observed symmetric CH$_4$ signals, neither in amplitude nor for its systematic occurrence. Figure 3 shows XCH$_4$ as a function of time and SZA for two exemplary days of standard TCCON data with and without AMD from the Ny-Ålesund site.





We define the AMD as the slope of the linear function fitted to the XCH$_4$-SZA data within a day. The AMD has units of ppb per degree and is only calculated for days with a sufficient range of SZAs ($> 5\,^\circ$ coverage).

To investigate whether the AMD is related to the presence of the vortex, the daily mean XHF can be used as a tracer for vortex air. Figure 4 shows the AMD as function of XHF for the four TCCON stations. Data between January and May are highlighted, as these are the months when CH$_4$-depleted air can be measured. Data between June and December are included for reference, as the descent of stratospheric air inside the vortex needs time to manifest. Additionally, we highlight data points which lie inside the polar vortex using our vortex mask (see Sec. 3.1).

Data between June and December have a mean AMD value close to zero with values between $-0.18\,\mathrm{ppb\,deg^{-1}}$ and $0.17\,\mathrm{ppb\,deg^{-1}}$ (change per SZA) for the four stations. During January and May, when the influence of the vortex is expected, the mean values increase to $0.68 - 1.29\,\mathrm{ppb\,deg^{-1}}$. Data clearly identified as inside-vortex have mean values from $1.26 - 3.59\,\mathrm{ppb\,deg^{-1}}$.

A clear tendency of higher AMD for higher XHF (and hence inside-vortex air) can be seen, which is also indicated by the linear fits shown in Fig. 4. A stronger descent of stratospheric air inside the vortex leads to higher XHF and thus less XCH$_4$. A high XHF values is thus correlated to a more "depleted" CH$_4$ profile shape. It can be seen that more depleted air leads on-average to higher AMD. This relation only holds true to a certain extent, as indicated by the wide spread of AMD and the relatively low Pearson correlation coefficients of $\rho = 0.15$ to $0.48$. This can be explained by a) other effects causing AMD, which have not been corrected by the airmass-dependent correction factor and are not considered here, b) the existing prior not being consistently wrong (the difference between prior and true profile shape can vary) or c) true changes in diurnal XCH$_4$ caused by local emissions or changes in atmospheric transport.

### 4.1 Impact of wrong prior shape

Here we briefly explain how the wrong prior profile shape leads to AMDs. The TCCON retrieval is a least-square profile scaling retrieval (PSR). During the retrieval, a scaling factor $\hat{\gamma}$ is determined, which is used to scale the prior profile $\mathbf{x_a}$ to minimize the cost function. The retrieved total column $\hat{c}$ is then simply given by the scaled prior total column

$$\hat{c} = \hat{\gamma}\mathbf{h}^T\mathbf{x_a} = \hat{\gamma}c_a \tag{1}$$

where $\mathbf{h}^T$ is the vector of pressure weights and $\mathbf{h}^T\mathbf{x_a}$ the dot product between both vectors.

Following Rodgers (2000) and Rodgers and Connor (2003), the retrieval can be approximated as an estimate of the true state smoothed by the averaging kernel

$$\hat{c} = \hat{\gamma}c_a + \mathbf{h}^T\mathbf{A}(\mathbf{x} - \hat{\gamma}\mathbf{x_a}) + \epsilon \tag{2}$$

where $\mathbf{A}$ is the averaging kernel and $\epsilon$ the summary of all additional error terms (e.g., retrieval noise), which we do not consider here. The averaging kernel matrix describes the sensitivity of the retrieval to the true state $\frac{\partial \hat{x}}{\partial x}|_{x=\gamma x_a}$ evaluated at the linearization point $x = \gamma x_a$. Note that the TCCON averaging kernels are calculated for the retrieved value $\hat{x} = \hat{\gamma}x_a$. Since TCCON uses a PSR, this matrix only contains $N$ pieces of information ($N$ is the number of altitude levels). The column





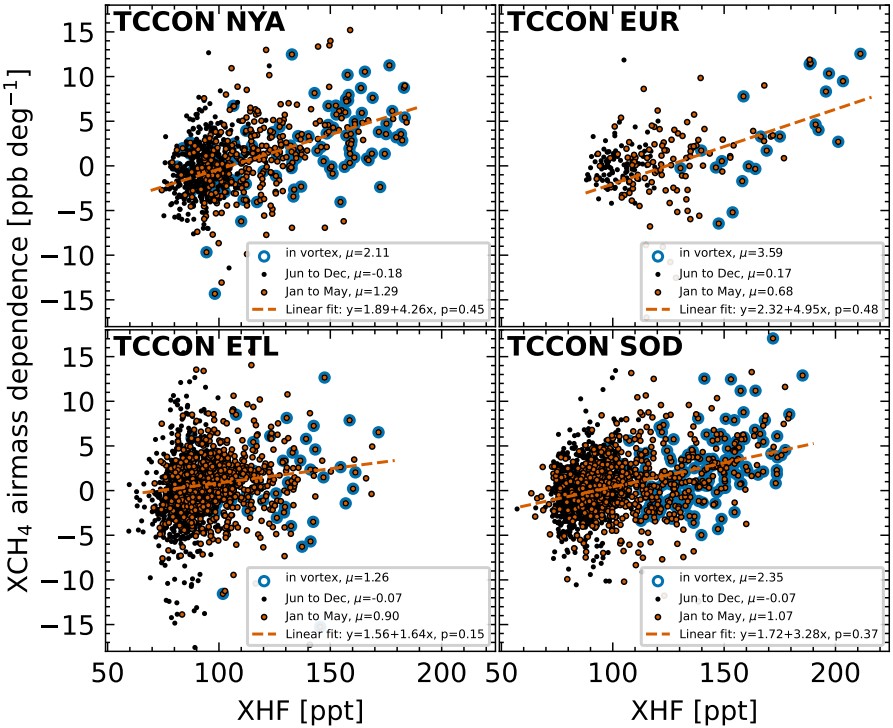

**Figure 4.** Daily AMD as a function of daily averaged XHF for the four Arctic stations. The AMD is defined as the slope of a linear function fit between XCH$_4$ and the solar zenith angles for each day. Mean values ($\mu$) are provided for data flagged as inside-vortex, for January to May and June to December. Additionally, linear fits are calculated from January to May; $\rho$ is the Pearson correlation coefficient.

averaging kernel $\mathbf{a}^T$ is a vector $\mathbf{a} = \frac{\partial \hat{c}}{\partial \mathbf{x}}\big|_{x=\gamma x_a}$, which implies that changes in the true profile lead to equal changes in the
165 retrieved total column (since only the scaling factor can be adjusted which scales all levels equally). The averaging kernel is mainly dependent on the prior profile and airmass and consequently on the solar zenith angle.

During the polar spring, SZAs are still relatively large, especially in the morning and evening. Figure 5 shows the NYA column averaging kernels for a single day in April 2020. For large SZA measurements, the AK is around 0.6 in the stratosphere and around 1.2 in the troposphere.

Returning to Eq. 2 we can now explain part of the observed AMDs. During measurements under high SZA conditions, the averaging kernel is small in the stratosphere and large in the troposphere. This is important, since smaller AK values mean that more information comes from the prior profile (in an idealized measurement with $\mathbf{A} = \mathbf{1}$ all information comes from the measurement). If the shape of this prior profile is sufficiently wrong, this leads to AK- and hence SZA-dependent artifacts in the XCH$_4$ (an airmass dependence), which are proportional to the difference between the true and prior CH$_4$ profile.





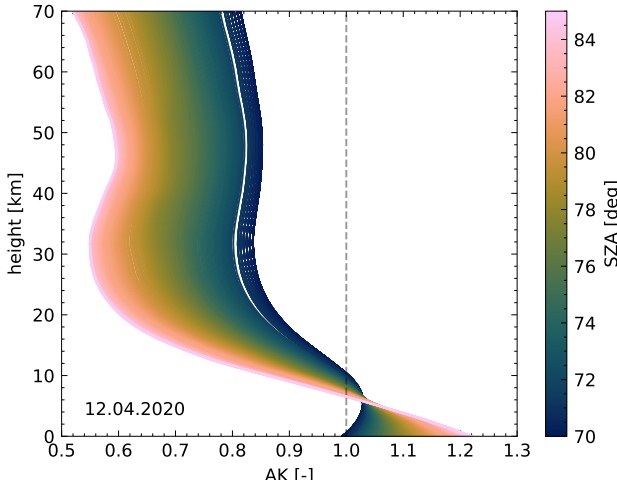

**Figure 5.** TCCON Ny-Ålesund XCH$_4$ averaging kernel for the 12 April 2020. Colors show the dependence on the solar zenith angle. For high angles the averaging kernel is less sensitive to the stratosphere and overly sensitive to the troposphere.

## 5 CH$_4$ profiles in polar vortex

In the previous section, we investigated the AMD present in Arctic TCCON XCH$_4$ and suggested wrong prior shapes as a possible explanation. This section presents a brief overview of CH$_4$ profile measurements from different sources and a comparison to the TCCON CH$_4$ priors. Qualitatively similar differences can be observed between various measurements shown below and the TCCON priors, indicating that systematic errors are present in the TCCON CH$_4$ prior.

### 5.1 ACE-FTS

The Atmospheric Chemistry Experiment Fourier Transform Spectrometer (ACE-FTS) is the main instrument onboard SCISAT-1 and provides profile measurements of various atmospheric species (Bernath, 2017). Here we analyze CH$_4$ profiles inside and outside the polar vortex. The measurements are collocated using the latitude and longitude provided by the ACE-FTS files, which correspond to the 30 km geometric tangent point. We analyzed profiles above 50 °N between January and May for the years 2009–2022 from ACE-FTSv5.2 data (Boone et al., 2023). Figure 6 shows the mean in- and out-of-vortex CH$_4$ profiles in comparison to the mean inside-vortex TCCON prior.

### 5.2 NDACC

The Network for the Detection of Atmospheric Composition Change (NDACC) is a global network of ground-based remote-sensing stations that employ various instruments to measure the atmospheric composition. Here, we are interested in CH$_4$ profiles retrieved from solar absorption FTIR measurements in the mid-infrared in Ny-Ålesund and Eureka. To enable direct comparison between NDACC profiles and TCCON priors (see Sec. 5.4), the closest TCCON measurement within a day was

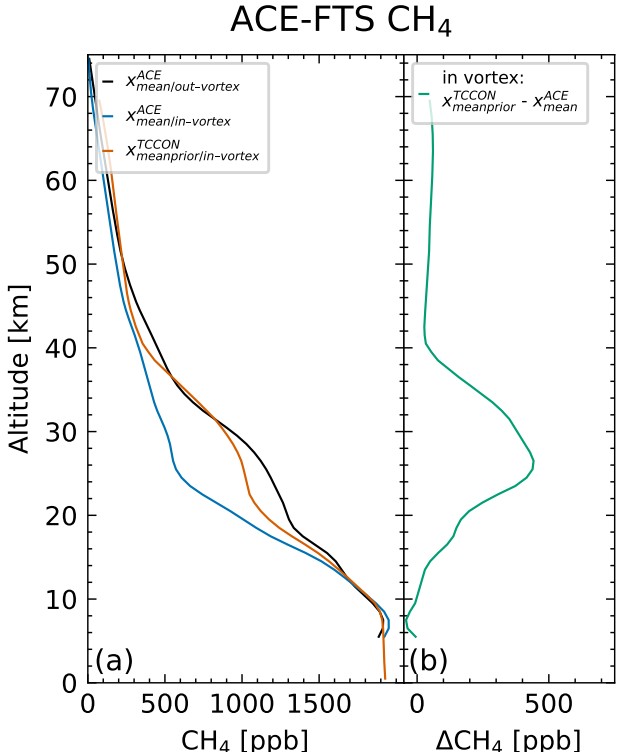

**Figure 6.** (a) Mean in- and out-of-vortex $CH_4$ profiles from ACE-FTS measurements and mean inside-vortex standard TCCON $CH_4$ prior (averaged over NYA, SOD, EUR, ETL). For the calculation of mean profiles, measurements between 2009 and 2022, January to May, north of 50 °N were used. Panel (b) shows the difference between the mean TCCON prior and mean ACE-FTS profile inside the vortex.

collocated to each NDACC measurement. Panel (a) in Fig. 7 shows the mean in- and out-of-vortex NDACC profiles and the corresponding mean inside-vortex TCCON priors.

### 5.3 AirCore

AirCore is a technique used to measure vertical profiles of atmospheric gases, including $CH_4$, carbon dioxide ($CO_2$), and others (Karion et al., 2010). The method involves lifting an AirCore coil of specifically coated and filled tubing by a meteorological balloon to altitudes of 30-35 km, well above typical aircraft flight levels. As the payload descends, the AirCore coil, open at one end, fills with air, which is then analyzed within a few hours of landing. At Sodankylä, a Picarro G2401 cavity ring-down spectrometer is used to analyze the air samples. The instrument measures $CO_2$, $CH_4$, and CO with precision and accuracy of

0.05 and 0.1 ppm, 0.5 and 1 ppb, and 8 and 3 ppb respectively. By calibrating the analyzer with gas standards traceable to WMO scales, absolute trace gas mole fractions are determined.

Due to the complicated logistics - the AirCore needs to be retrieved and analyzed within a short window of time - only a limited number of AirCores are available during polar vortex conditions. Only two AirCores between 2017–2021 show distinct





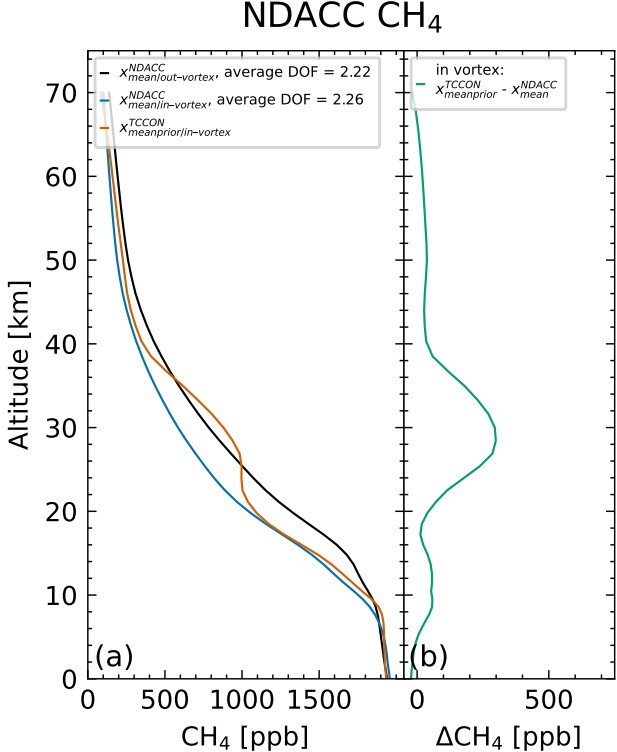

**Figure 7.** (a) Mean in- and out-of-vortex NDACC $CH_4$ profiles and mean inside-vortex standard TCCON $CH_4$ prior (averaged over NYA and ETL). Panel (b) shows the mean difference between the TCCON prior and NDACC profile inside the vortex.

inside-vortex $CH_4$ profiles, with reduced $CH_4$ concentrations in the stratosphere. The profiles were obtained two days apart

and lie within the vortex region during its dissolution. Figure 8 shows these two profiles in comparison to the corresponding TCCON prior and a reference summer AirCore.

### 5.4 TCCON prior

The TCCON prior profiles are generated using the "ginput" algorithm (Laughner, 2022) with the Goddard Earth Observing System Forward Product for Instrument Teams (GEOS FP-IT) reanalysis product as input. The calculation of the TCCON prior

profiles is described in Laughner et al. (2023). The stratospheric prior is based upon the work of Andrews et al. (2001), which showed that profiles of $CO_2$ and $N_2O$ in the lower stratosphere are captured well by in situ observations from Mauna Loa and American Samoa. First, the age-of-air is calculated from a climatology simulated by the Chemical Lagrangian Model of the Stratosphere (McKenna et al., 2002a, b). The stratospheric profile is then constructed by using the age-of-air to determine the the mole fraction of each gas a parcel of air had when entering the stratosphere. The mole fractions are taken from the

averaged Mauna Loa and American Samoa time series lagged by two months. Additionally, ACE-FTS data are used to account for chemical production and loss of chemically active gases in the stratosphere.



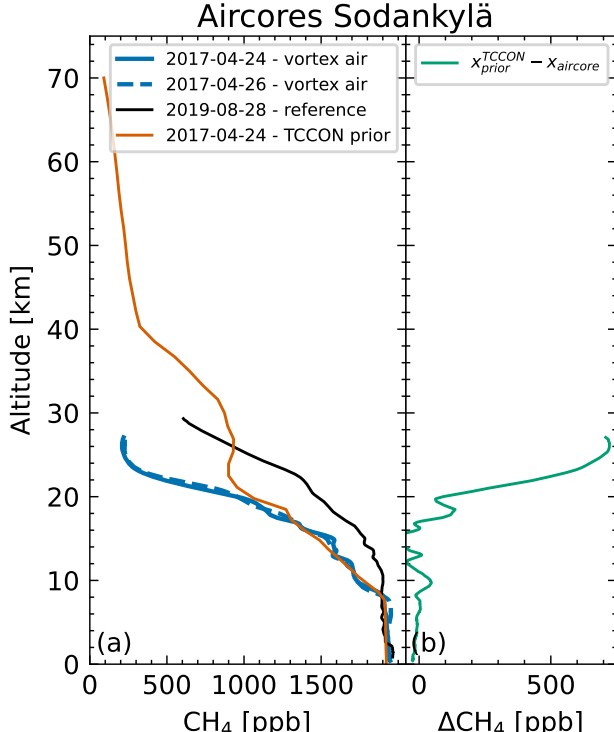

**Figure 8.** (a) $CH_4$ profiles from AirCores at Sodankylä. The two AirCores measuring vortex air, show a distinct reduction of $CH_4$ in the upper atmosphere compared to out-of-vortex AirCores (reference), measuring at a similar time of the year. Additionally, significant differences are visible between AirCore profile and corresponding TCCON prior profile. The difference between both is shown in panel (b).

Figure 9 shows differences between average TCCON prior profiles inside and outside the polar vortex for the four Arctic TCCON sites. A clear difference between in- and out-of-vortex profiles is visible between 10 km and 30 km, with maximum differences at roughly 20 km. However, comparison of TCCON prior profiles to measured profiles indicate that TCCON

priors underestimate the stratospheric inside-vortex $CH_4$ decrease. Differences up to 700 ppb are visible in panel (b) of Fig. 8, comparing the TCCON prior to an AirCore profile. The difference is largest around 26 km and roughly follows a partial Gaussian curve, decreasing with reduced altitude. Panel (b) in Fig. 6 shows differences between the mean inside-vortex ACE-FTS profile and mean inside-vortex TCCON prior profile. For the latter, inside-vortex priors for all four stations (NYA, SOD, ETL, EUR) were averaged; for the former we averaged all inside-vortex profiles between January and May, 2009–2022 and

north of 50 °N. Differences up to 400 ppb are visible with a maximum at roughly 26 km. The shape of the difference roughly agrees with Fig. 8, with a maximum difference around 26 km and decreasing difference with decreasing altitude. Lastly, we collocated inside-vortex TCCON priors and NDACC profiles for Eureka and Ny-Ålesund and calculated the mean difference between them which is visible in panel (b) of Fig. 7. The differences between TCCON and NDACC show a similar Gaussian shape as the differences between TCCON and ACE-FTS or the AirCores, albeit with a lower magnitude of roughly 250 ppb. A



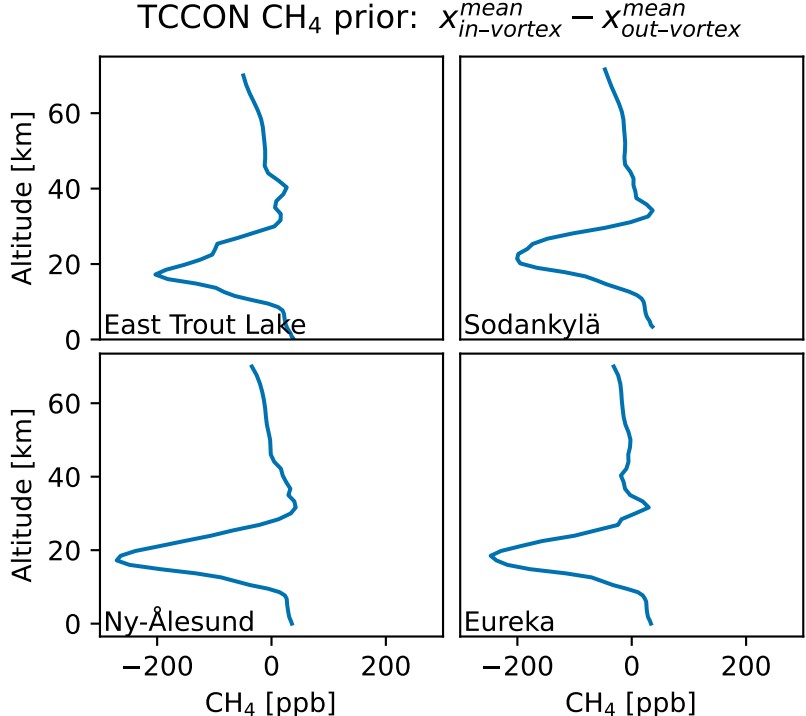

**Figure 9.** Difference between in- and out-of-vortex TCCON $CH_4$ prior profiles. Profiles are classified as in- and out-of-vortex using a polar vortex mask derived from ERA5 data described in 3.1.

possible explanation for this, could be the strong regularization of the NDACC retrieval which limits the amount of change in the profile shape. The NDACC $CH_4$ retrieval has 2 degrees of freedom for signal, which allows separation of the tropospheric and stratospheric columns. Dependence on the prior is reported to be typically around 1% (Bader et al., 2017).

The opportunities for comparing TCCON prior profiles to measurements are limited. The comparison between the TCCON prior and AirCore profiles demonstrated the difference for a single example of inside-vortex measurement. Due to the limited

co-location between ACE-FTS and TCCON measurements, we analyzed differences between averaged inside-vortex priors and profiles in Fig. 6. The difference between the TCCON prior shape and profile measurements are systematic. The disagreement is largest at roughly 26 km and decreases roughly symmetrically around this height. Similar results, with a smaller magnitude, can be seen in the comparison between NDACC profiles and TCCON priors. In Sec. 6, the observations are used to inform the prior modification.





## 6 TCCON prior modification

Three modifications of the TCCON $CH_4$ prior have been tested and are briefly described in the following sections. We constrain our analysis to data at the beginning of the year from January through May, since only in early spring, when first measurements are available, the descent is sufficiently visible in the data (recalling that the vortex only fully manifests by mid-November).

### 6.1 Static prior

The static modification was the first attempt to improve the TCCON $CH_4$ prior. It was created and optimized for inside-vortex NYA data (by testing various configurations) and subsequently applied to ETL and SOD data. The static prior was informed by the observations described in Sec. 5. For the sake of simplicity and to reduce the number of free parameters, we approximated the differences by a Gaussian function. The priors are then modified by multiplying with the (shifted) Gaussian function, leading to a reduction of stratospheric $CH_4$ while keeping the rest of the profile shape unchanged. A detailed description on how the static modification is done can be found in Appendix B. One disadvantage of this approach is the need for some additional criterion on when to apply the modification. A possible choice would be a polar vortex mask as described in Sec. 3.1. Here we use XHF>100 ppt as a criterion, to ease comparability with the dynamic prior.

### 6.2 Dynamic prior

The dynamic modification improves on the static modification by scaling the change to the prior with the TCCON XHF. This is based on the assumption that XHF is sufficiently correlated to the strength of down welling in the stratosphere, which affects the $CH_4$ profile shape. No vortex mask is needed in this case. The magnitude of modification is linearly scaled between 0 and 1 from 100 ppt to 180 ppt. Priors for which XHF is below the threshold are not modified and the maximum modification is reached at 180 ppt after which the maximum modification is used. Similar to the static prior, this modification was first created and optimized for NYA. A detailed description on how the dynamic modification is done can be found in Appendix C.

### 6.3 Model prior

The model prior modification uses profile information from the TCOM-$CH_4$ dataset, which combines TOMCAT/SLIMCAT model data with ACE-FTS satellite observations (Dhomse and Chipperfield, 2023). The stratospheric TCCON prior is replaced by the model profile between roughly 12 km and 61 km for all spectra. This modification was tested for NYA data, to check whether model data could further improve upon the relatively simple dynamic prior.

## 7 Results

Retrievals using modified priors were performed for NYA, SOD, ETL and EUR. Retrievals using the static priors were performed for NYA, SOD and ETL. Retrievals using the dynamic prior were performed for all three stations. The model prior was only tested for NYA.





First, we present changes in the AMD when using different priors. Next, we present the changes in the spectral fits, quantified
by the root-mean-square of the fitting residuals. Lastly, we show the changes in averaging kernels for a subset of NYA spectra.

## 7.1 Airmass dependence

Figures 10, 12, 13 and 14 present the $XCH_4$ AMD as function of XHF for the standard prior as well as the static and dynamic
prior for NYA, SOD, ETL and EUR, respectively. We calculate the mean and standard deviation for four regions, to investigate
the AMD-XHF dependence. Here no linear fits (as in Fig. 3) are used, because the amount of data varies between stations (e.g.,
few high XHF data for ETL) and thus linear fits do not provide a robust tool for investigation. Measurements that lie within the
vortex mask are marked in the plot to provide additional information.

For NYA, an increasing bias can be observed with values up to $\mu = 4.82$ ppb deg$^{-1}$. The static prior was especially designed
for inside-vortex measurements and thus yields a significant bias for high-XHF measurements (which are mostly flagged as
inside-vortex). For XHF$< 140$ ppt, the bias is, however, significantly increased to negative values. The dynamic prior was
designed for measurements with XHF$> 100$ ppt and leads to an overall improvement with values below $\mu = 1.06$ ppb deg$^{-1}$.
For NYA, we also tested the model prior as shown in Fig. 11; the improvement is similar to that of the dynamic prior. The
corresponding standard deviations vary little between the different priors, suggesting that AMDs independent of the polar
vortex are present in the data, which stay mostly unchanged by our modifications.

For SOD, similar improvements as for NYA are visible. The standard prior exhibits a strong bias for XHF$> 140$ ppt. The
static prior reduces these biases significantly while introducing a negative bias to data with XHF$< 140$ ppt. The dynamic prior
again leads to an overall improvement. Only the absolute bias between 120 and 140 ppt slightly increases compared to the
standard prior.

For ETL, the standard prior exhibits smaller biases compared the other stations. For XHF$> 140$ ppt, the static prior again
leads to a reduction of the absolute bias. However, the improvement is smaller compared to NYA and SOD and the sign change
in the bias indicates the static prior modification over corrects the prior in this case. The dynamic prior leads to an overall
improvement, similar to that for SOD, while the absolute bias between 120 and 140 ppt increases compared to the standard
prior.

For EUR, only the dynamic prior is available. Significant improvements are visible for high XHF$> 140$ ppt and for values
between 120 and 140 ppt. For the other two regions, biases are slightly increased. Part of this mixed picture is likely a result
of the limited data available.

Overall, the dynamic prior reduces the average AMD for most data for all four stations. For NYA, the dynamic prior shows
the best results, while for SOD and ETL over corrections are visible for the range $140 >$XHF$\geq 120$ ppt. For NYA, the model
prior exhibits a similar improvement to the dynamic prior. The static prior reduces the mean AMD for XHF$> 140$ ppt for all
three stations, while it introduces negative biases for XHF$< 140$ ppt. For EUR only the dynamic prior is available. Results are
mixed (possibly due to the smaller amount of available data) but significant improvements are visible for XHF$> 140$ ppt.



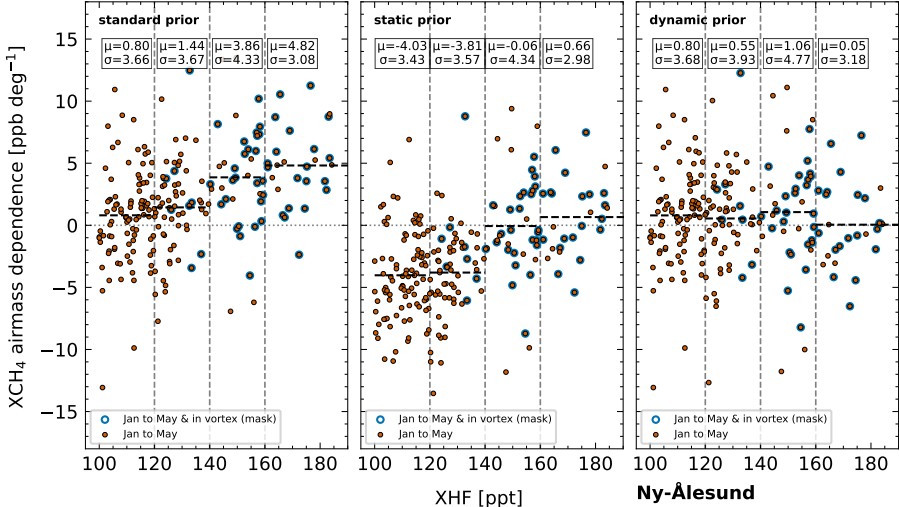

**Figure 10.** Airmass dependence as function of XHF for different $CH_4$ priors at Ny-Ålesund. $\mu$ is the mean AMD and $\sigma$ the standard deviation of AMD for a given region defined by the dashed gray vertical lines.

## 7.2 Spectral fits

The root-mean-square (RMS) of the spectral fitting residuals is the standard deviation of the difference between the measured and calculated spectra. It can be used to compare retrievals with different priors. If the prior shape is closer to the true profile shape, the calculated spectra will be closer to the measured spectra and the RMS will be smaller. Conversely, a wrong prior
shape can lead to larger RMS. To make the RMS comparable between different signal levels, it is divided by the continuum level. The resulting measure is called RMS/CL and is separately calculated for each fitting window. In the following, we will analyze the relative improvement in RMS/CL compared to the standard retrieval $\Delta_R$. Positive values of $\Delta_R$ constitute an improvement of the fit (lower RMS/CL), negative values an increase in RMS/CL compared to the reference retrieval.

First, we investigate the improvement in spectral fits for NYA for which all three different prior modifications are available.
Figure 15 shows the improvement in RMS/CL compared to the standard retrieval ($\Delta_R$) for all three modified priors and fitting windows using a 2D histogram. The color bar is logarithmically scaled, displaying the Number $N$ of spectra per bin. Additionally, we show the percentage $n$ of spectra for the upper half (improvement of fit) and lower half (decline of fit) of each panel. Note that we only investigate spectra with XHF>100ppt, which is the lower cutoff value for modification with the dynamic prior and static prior. The largest differences can be seen for the window centered on $6002 \ \mathrm{cm}^{-1}$, which includes
the Q-branch of the $CH_4$ absorption band (middle column), on which we focus for now. For the static prior, the $\Delta_R$ declines up to 25% for most XHF values below 150 ppt and increases by 20% for values above 150 ppt. The dynamic prior shows improvements of $\Delta_R$ increasing with XHF and maximum improvements of 25% at roughly 175 ppt. However, some spectra with $\Delta_R$ reduced by up to 10% are also present. The other fit windows show similar results with a lower range of $\Delta_R$ between -5% and 7%.





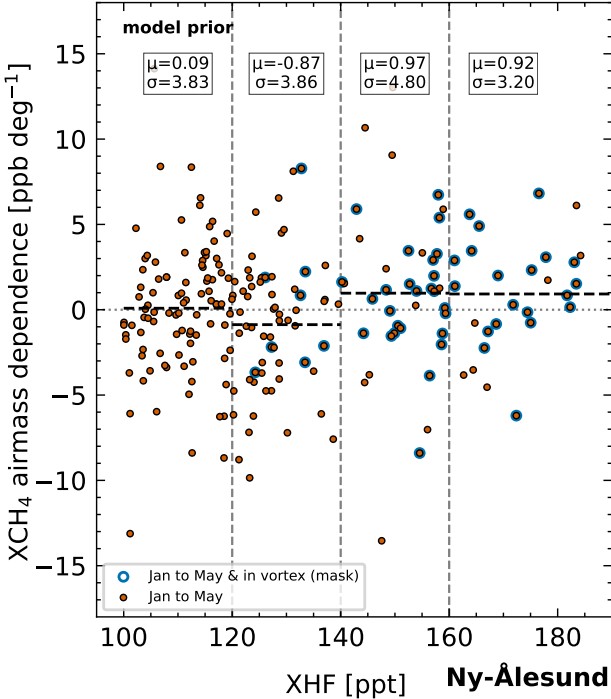

**Figure 11.** Airmass dependence as function of XHF for different $CH_4$ priors at Ny-Ålesund. $\mu$ is the mean AMD and $\sigma$ the standard deviation of AMD for a given region defined by the dashed gray vertical lines.

Overall roughly 90% of measurements have improved $\Delta_R$ (i.e., lower RMS/CL) across all three windows when using the dynamic prior. For the static prior, roughly 70% of measurements have a reduced $\Delta_R$ (i.e., higher RMS/CL) compared to the standard retrieval.

Figure 16 shows the changes in $\Delta_R$ for SOD. An overall decline in $\Delta_R$ is visible in all fit windows for the static prior, compared to the standard retrieval. The dynamic prior shows an overall improvement in all fit windows. Improvements are

between roughly 53% and 72% for the different fit windows and thus smaller than for NYA. Still, the RMS/CL improves for the majority of spectra when using the dynamic prior for SOD.

Figure 17 shows the results for ETL. ETL exhibits similar results as NYA and SOD for the static prior, with a worsening of spectral fits for most spectra with XHF below $150$ ppt and improvements for high XHF values above $150$ ppt. The dynamic prior achieves an improvement in the majority of spectra for all three windows, as observed for NYA. However, the

improvements are smaller than for NYA, ranging from 50% to 70%.

Figure 18 shows the results for EUR. Only results for the dynamic prior are available, with overall improvements in all fit windows. Improvements are between roughly 54% and 67%, and thus smaller than for NYA but comparable to SOD.




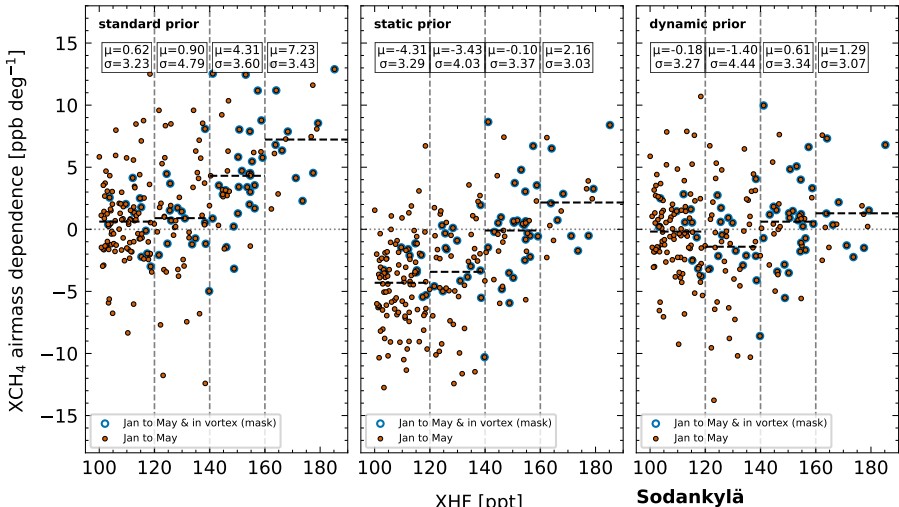

**Figure 12.** Airmass dependence as function of XHF for different $CH_4$ priors at Sodankylä. $\mu$ is the mean AMD and $\sigma$ the standard deviation of AMD for a given region defined by the dashed gray vertical lines.

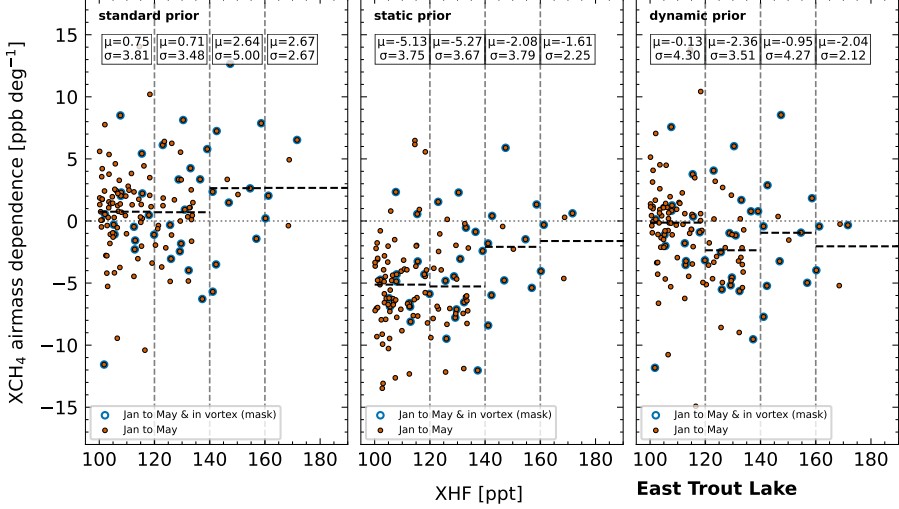

**Figure 13.** Airmass dependence as function of XHF for different $CH_4$ priors at East Trout Lake. $\mu$ is the mean AMD and $\sigma$ the standard deviation of AMD for a given region defined by the dashed gray vertical lines.

## 7.3 Averaging kernels

The averaging kernels are an important part of the retrieval results and provide information on the sensitivity of the retrieval to the true atmosphere. The averaging kernels depend on the sensitivity of the retrieval method and observing system to the state as well as the sensitivity of the forward model to the state. Usually the SZA is the main influence on the averaging kernels.






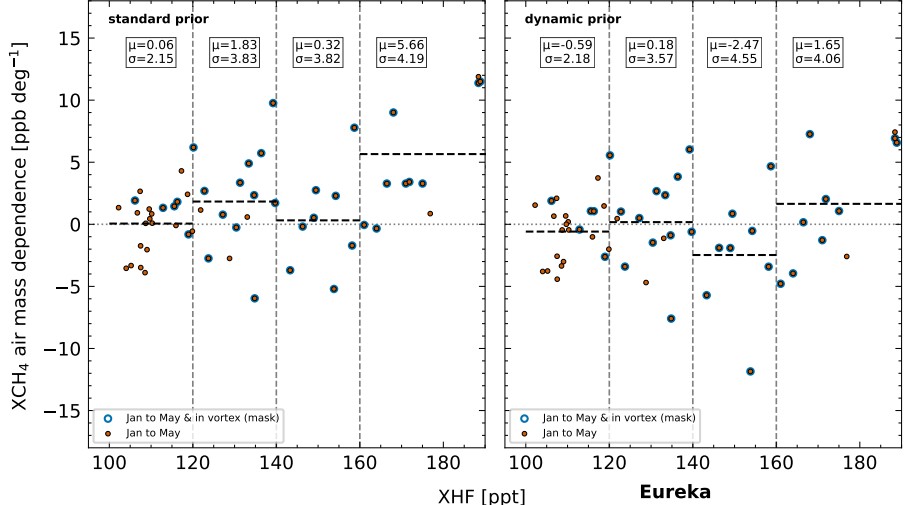

**Figure 14.** Airmass dependence as function of XHF for different $CH_4$ priors at Eureka. $\mu$ is the mean AMD and $\sigma$ the standard deviation of AMD for a given region defined by the dashed gray vertical lines.

However, the averaging kernels are technically dependent on the prior since they are calculated for the scaled prior. Here we calculate the AKs for a subset of data and compare AKs between the standard retrieval and modified retrieval. Due to the large amount of storage necessary to calculate the AKs, we confine our analysis to the NYA site, the dynamic prior, and a sample of

1000 spectra with XHF> 100 ppt.

Figure 19 shows the AKs for 1000 randomly sampled spectra from NYA with XHF> 100 ppt between January and May using the standard prior. The 6002 and 6076 $\text{cm}^{-1}$ windows show a similar shape, with values below 1 in the stratosphere and values above 1 in the troposphere. However, the 6002 $\text{cm}^{-1}$ window shows a stronger deviation from 1, with values down to 0.3 in the stratosphere compared to 0.6 for the 6067 $\text{cm}^{-1}$ window. The 5938 $\text{cm}^{-1}$ window shows a similar shape for SZAs

larger than roughly 77 °. Below this threshold, the AK is above 1 in the stratosphere and below 1 in the troposphere. The deviation from 1 is relatively small in this window, with values ranging between 0.8–1.2. In general, the AK is increasingly deviating from 1 for high SZAs, as we have already seen in Fig. 5.

Figure 20 shows the relative change in AKs when using the dynamic prior compared to the standard prior. In general, the AKs are improving as the values increase by up to 2.5% in the stratosphere and decrease by up to 1% in the troposphere,

hence bringing the AKs overall closer to 1. The changes are strongest in the 6002 $\text{cm}^{-1}$ window, followed by the 6076 $\text{cm}^{-1}$ window. Changes in the 5938 $\text{cm}^{-1}$ window are below 1%.

To assess the influence of the changed AK, we calculate the absolute change $\Delta\text{XCH}_4$ using a fixed example profile $x_i$ and pressure weights $h_i$.

$$\Delta\text{XCH}_4 = \sum_i \left( h_i \Delta A_i x_i \right) \qquad (3)$$





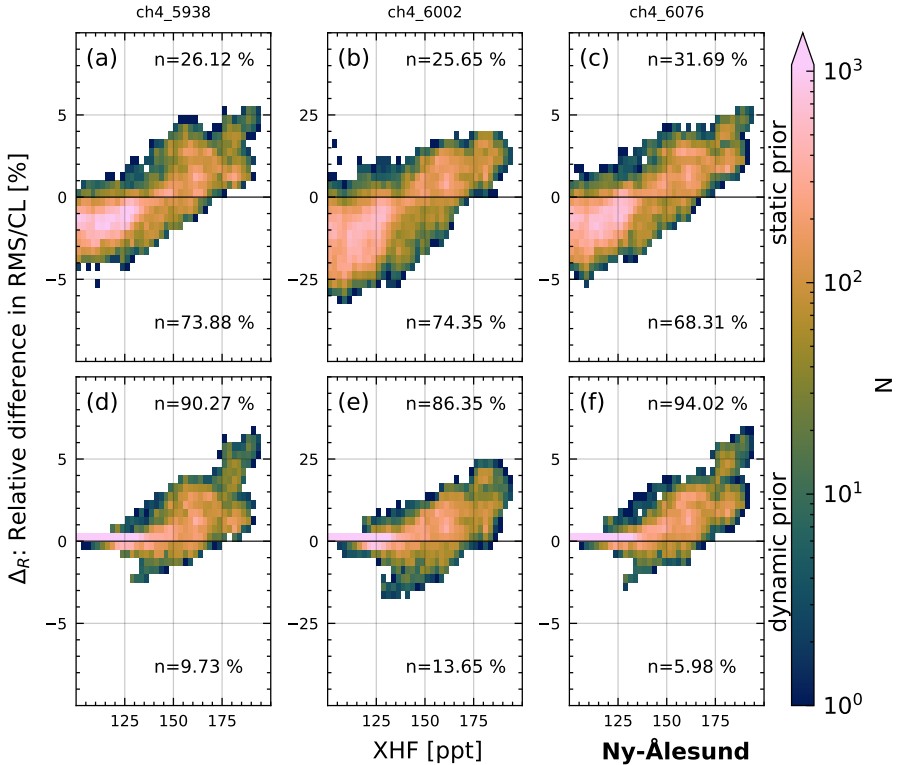

**Figure 15.** Histograms showing the improvement in spectral fits as a function of XHF for different priors (rows) and the different CH$_4$ windows (columns) for NYA. The improvement in spectral fit $\Delta_R = 100\% \times (RMS/CL_{standard} - RMS/CL_{modified})/RMS/CL_{standard}$ is given as the relative difference between the RMS/CL (root mean square divided by continuum level signal) between the standard retrieval and modified retrieval. The color bar is logarithmically scaled, displaying the Number $N$ of spectra per bin. $n$ are the percentage of spectra with improved/reduced $\Delta_R$ and are shown in the upper and lower half of each panel.

where $\Delta A_i$ is the relative change of the AK. This yields differences up to 10 ppb in magnitude and a mean difference of roughly 3.5 ppb.

### 7.4    AirCore comparison

Previous results were confined to the analysis of relative improvements between different versions of the TCCON retrieval. The comparison to the AirCores allows for the external validation of the previous results although the number of suitable AirCores

is limited. Two AirCores are available during vortex conditions, however only one AirCore coincides closely with a TCCON measurement. For the other AirCore, only one TCCON measurement within 3 hours is available, which is over 2 hours earlier than the AirCore launch. Thus, we analyze only the AirCore on the 24 April 2017 which closely coincides with TCCON measurements. To calculate the column-averaged dry-air mole fraction, the AirCore profile was interpolated to the TCCON pressure grid and extended using the standard TCCON prior, which yields a value of $c_{AirCore} = 1768.22$ ppb. We additionally



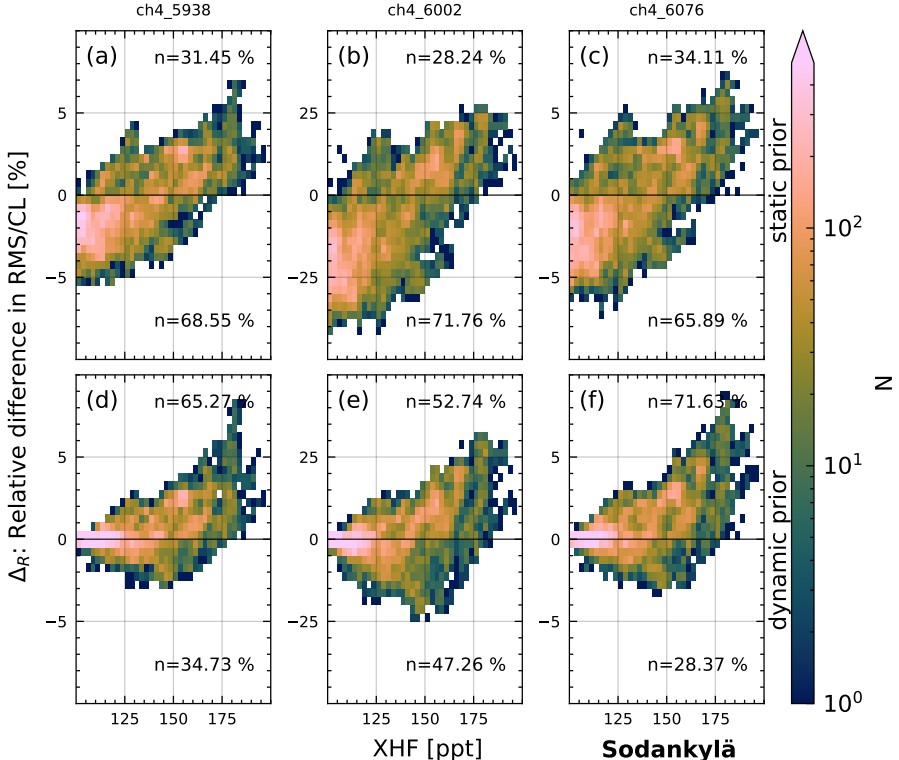

**Figure 16.** Same as Fig. 15, but for SOD.

performed a retrieval using the extended AirCore profile as the prior. This retrieval should constitute the best possible retrieval
of the spectrum given the available information, as the prior profile should closely match the true profile.

Usually, TCCON measurements are compared to smoothed AirCore (or aircraft) measurements following the approach of
Wunch et al. (2010). We call the difference between TCCON and the smoothed in-situ $XCH_4$ the smoothed bias. This approach
is used to determine an airmass-independent correction factor (AICF), to correct the systematic biases between TCCON and
the in situ measurement which is tied to the WMO scale (Wunch et al., 2010; Messerschmidt et al., 2011; Geibel et al., 2012).
Smoothing the AirCore profile using the different priors yields values of $c_{AirCore}^{standard} = 1772.55$ ppb, $c_{AirCore}^{static} = 1769.64$ ppb,
$c_{AirCore}^{dynamic} = 1769.65$ ppb and $c_{AirCore}^{AirCore} = 1768.21$ ppb. As expected, these values approach the "true" AirCore $XCH_4$ $c_{AirCore}$,
as the prior profile shape approaches the true profile shape. The differences $\Delta c = \hat{c}_{TCCON} - \tilde{c}_{AirCore}$ are between $0.85$ and
$1.85$ ppb. The smoothed bias is smallest for the dynamic prior and largest for the standard prior. We argue however that these
differences can't be meaningfully compared, as they depend on the averaging kernel, the difference between the measured air
masses, and additional systematic and random errors.



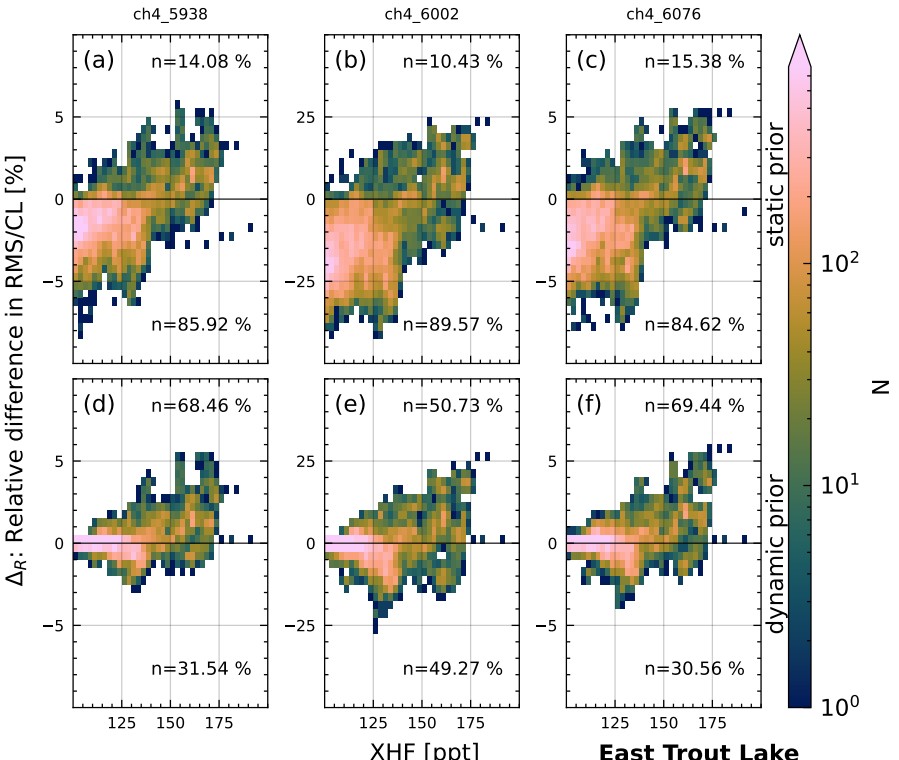

**Figure 17.** Same as Fig. 15, but for ETL.

The meaning of the difference $\Delta c$ can be clarified by substituting the retrieved TCCON column $\hat{c}_T$ with the linearized form of the retrieval (see Eq. 2)

$$\Delta c = \hat{c}_T - \tilde{c}_A \tag{4}$$

$$= \hat{c}_T - \left[ \hat{\gamma} c_a + \mathbf{h}^T \mathbf{A} (\mathbf{x_A} - \hat{\gamma} \mathbf{x_a}) + \epsilon_A \right] \tag{5}$$

$$= \left[ \hat{\gamma} c_a + \mathbf{h}^T \mathbf{A} (\mathbf{x} - \hat{\gamma} \mathbf{x_a}) + \epsilon_T \right] - \left[ \hat{\gamma} c_a + \mathbf{h}^T \mathbf{A} (\mathbf{x_A} - \hat{\gamma} \mathbf{x_a}) + \epsilon_A \right] \tag{6}$$

$$= \mathbf{h}^T \mathbf{A} (\mathbf{x} - \mathbf{x_A}) + \epsilon_T - \epsilon_A \tag{7}$$

where the subscript $T$ denotes the TCCON measurement, $A$ the AirCore and $a$ the TCCON prior. The last row shows that $\Delta c$ describes the difference between the true atmosphere as "seen" by TCCON $\mathbf{x}$ and as measured by the AirCore $\mathbf{x_A}$ as well as the error terms $\epsilon$. It can be usually assumed that $\mathbf{x} \equiv \mathbf{x_A}$, meaning the first part vanishes and only the differences in error terms remain. This allows to quantify retrieval errors using the smoothed bias.

However, we argue that the aforementioned assumption can not be confidently made when measuring during vortex conditons at the edge of the vortex (e.g., SOD) as the light path of the TCCON instrument with a high SZA can vary significantly from the path of the AirCore. Hence, $\Delta c$ describes the smoothed difference between $\mathbf{x}$ and $\mathbf{x_A}$ in addition to potential errors $\epsilon$.





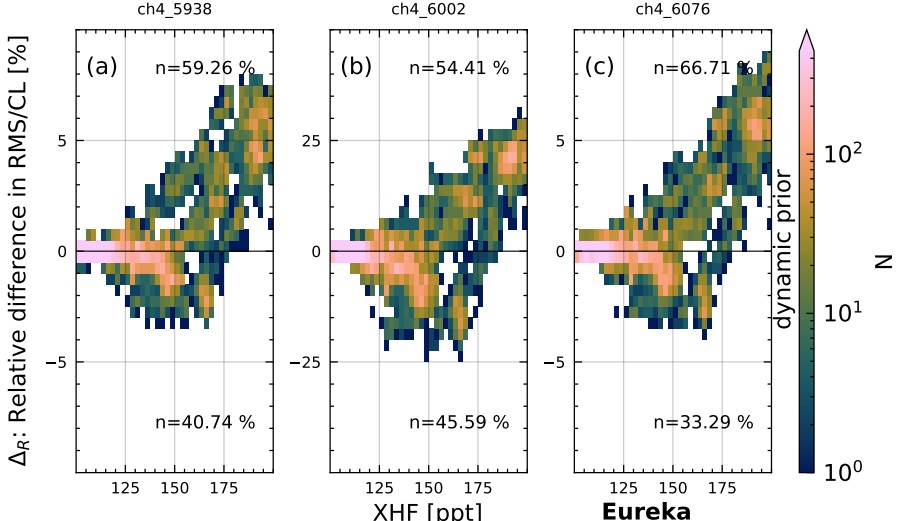

**Figure 18.** Same as Fig. 15, but for EUR.

Since in our case $\Delta c_i$ only vary within 1 ppb, we thus conclude that no further information can be deduced from the difference in $\Delta c_i$. In general, smoothed biases calculated for measurements with different averaging kernels can only be compared, if differences between $\mathbf{x}$ and $\mathbf{x_A}$ are expected to be minor in comparison to the retrieval errors.

Finally, Fig. 21 shows the scaled TCCON profiles next to the interpolated and extended AirCore profile. The standard TCCON profile has a clearly different shape from the AirCore above 20 km. To compensate for this wrong profile shape, the tropospheric $CH_4$ is underestimated compared to the AirCore profile (see inset plot). For the static and dynamic prior, the profile shape agrees better with the AirCore profile, although some differences are still visible in the stratosphere. The better agreement in profile shape is reflected in the better matching tropospheric $CH_4$ as well as the improved agreement between the dry-air mole fractions. When using the extended AirCore profile as prior information, even greater improvements can be observed, with the retrieval closely matching the AirCore profile.

## 7.5 Changes to TCCON XCH$_4$

Lastly, we want to briefly showcase changes to the XCH$_4$ between the standard and dynamic prior. Figure 22 shows the XCH$_4$ for the standard and dynamic prior as well as the difference $\Delta$XCH$_4$ for an exemplary time period. Differences up to roughly 12 ppb are visible, with $\Delta$XCH$_4$ exhibiting clear u-shapes, indicating an airmass dependence. This AMD in $\Delta$XCH$_4$ is precisely the AMD present in standard XCH$_4$ data that has been corrected in the XCH$_4$ using the dynamic prior. Maximum differences between standard and dynamic XCH$_4$ are roughly 17 ppb.



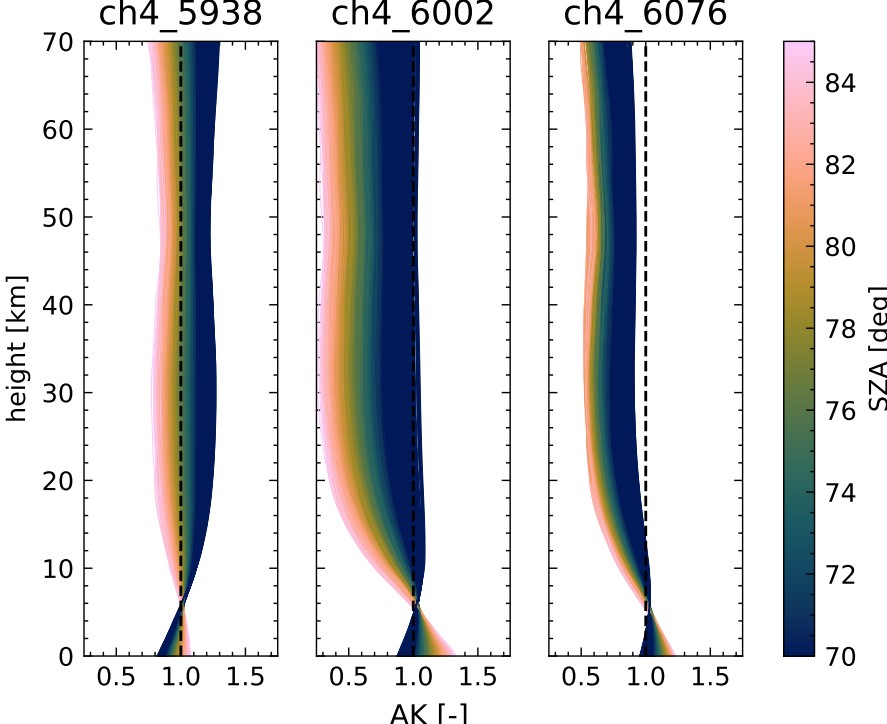

**Figure 19.** Averaging kernels for the three $CH_4$ fitting windows for the standard TCCON $CH_4$ retrieval for NYA. AKs were calculated for a subset of 1000 spectra randomly sampled from the all spectra with XHF$>$ 100 ppt between January to May.

## 8 Conclusions

In this paper, we investigated $XCH_4$ measurements from four Arctic TCCON stations during polar vortex conditions. We demonstrated that part of the existing airmass dependence (AMD) in TCCON $XCH_4$ is correlated to the presence of the polar vortex and that the AMD can be explained by the use of the wrong $CH_4$ prior profile shape. By improving the TCCON $CH_4$ prior shape to better represent inside-vortex conditions, various improvements could be noted: (i) AMDs in TCCON $XCH_4$ were reduced, (ii) spectral fitting residuals decreased, (iii) averaging kernels improved and (iv) the unsmoothed bias between TCCON and AirCore improved. While (i)–(iii) demonstrated relative improvements by comparing different TCCON retrievals, (iv) indicates that improvements in comparison to in-situ measurements are also possible. It should however be noted, that we only had one AirCore for direct comparison to TCCON data in vortex conditions, hence (iv) has only limited significance and demonstrates the need for more AirCore campaigns in the Arctic region that specifically target polar vortex conditions. Nonetheless, (i)–(iii) prove that improvements to the TCCON retrieval are possible using relatively simple modifications to the prior profile, which don't depend on external data. Improvements differed between the four sites Ny-Ålesund, Sodankylä, East Trout Lake, and Eureka. The largest improvements could be noted for Ny-Ålesund, which is the site for which the different prior modifications were first tested and designed. The same modifications work relatively well for Sodankylä, but it is evident



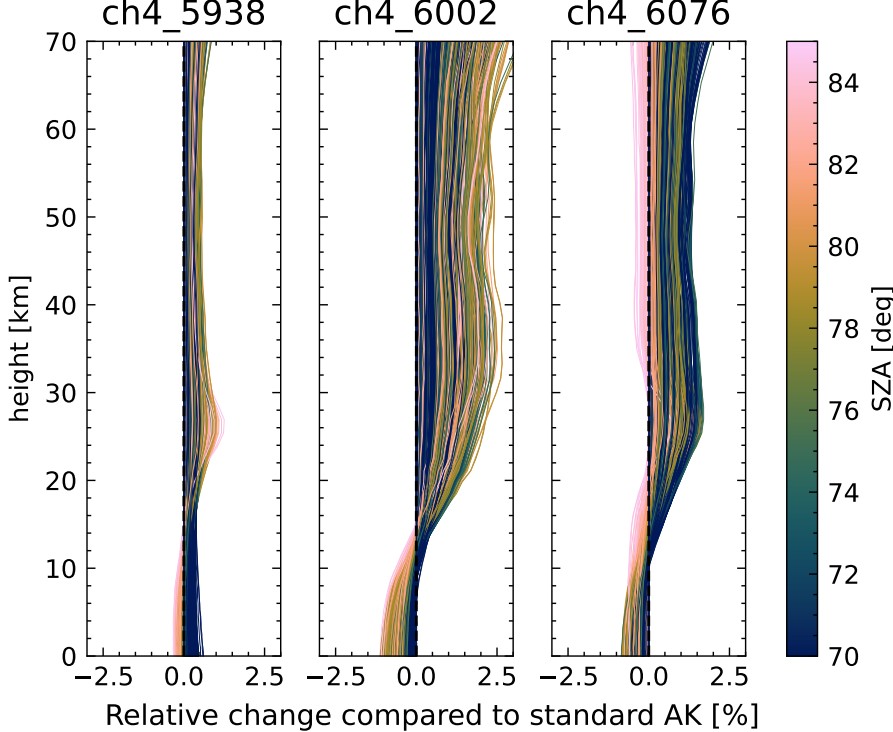

**Figure 20.** Relative change in averaging kernels between standard and dynamic prior TCCON CH$_4$ retrieval for NYA. The relative change is defined as $100\% \times (AK_{dyn} - AK_{ref})/AK_{ref}$. AKs were calculated for a subset of 1000 spectra randomly sampled from the all spectra with XHF$> 100$ ppt between January to May.

that a better prior modification could be designed for this site. For East Trout Lake some improvements could be observed, however the data suggests that the tested priors are not sufficiently optimized for this site. For Eureka improvements were mixed, with significant improvements only visible for high XHF. Interpretation of the results was further complicated by the smaller amount of data for that station.

These results are explicable considering the complex atmospheric processes involved. While Ny-Ålesund, Sodankylä and
Eureka are often situated inside the polar vortex, East Trout Lake typically only measures filaments of vortex air. In this case we expect the XHF and XCH$_4$ to be less correlated, as the XHF is quickly removed from the atmosphere when reaching the troposphere while XCH$_4$ experiences no such effect when transported from the lower stratosphere to the troposphere (or vice versa). As the air parcels have to travel hundreds or thousands of kilometers to reach East Trout Lake, it is likely that enough transport between the stratosphere and troposphere takes place to deteriorate the XHF-XCH$_4$ correlation used in our dynamic
prior.

In summary, we want to highlight that the prior shape has a significant impact on the retrieval (even in the stratosphere) and that considerable improvements are possible by changing the prior shape to better represent the true atmosphere, as the



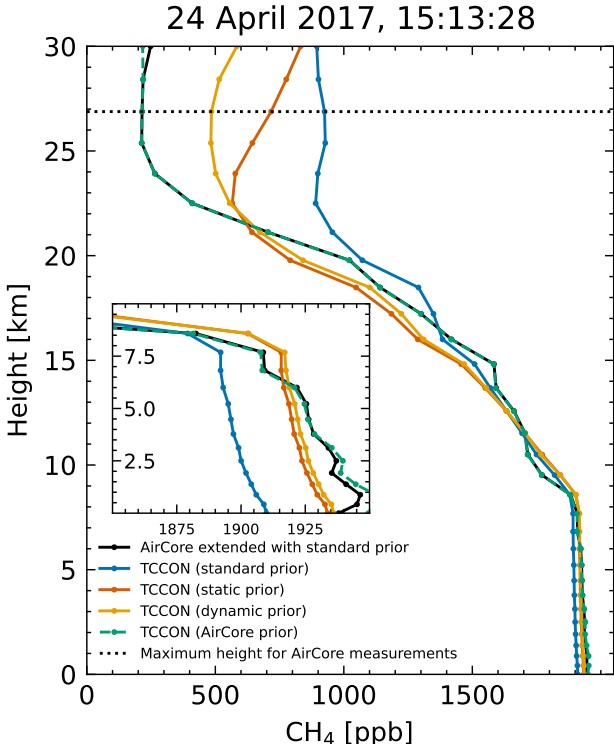

**Figure 21.** Comparison of scaled TCCON priors and the interpolated and extended AirCore profile for the Sodankylä site. The inset plot shows the lowermost 10 km where the improved agreement between the retrievals using the static or dynamic prior and the AirCore can be seen.

wrong prior shape can introduce systematic biases and AMDs. More in-situ measurements (i.e. AirCores) are therefore vital to improve our understanding of the Arctic atmosphere, which in turn allows the generation of more suitable prior profiles. We recommend that TCCON data during polar vortex conditions should be used with care and that the prior shape and averaging kernels (which are a vital part of the product) should be included in the analysis. Furthermore, we expect similar problems to be present in other profile-scaling retrievals and for other gases which exhibit a similar profile shape (e.g., $N_2O$).

*Code and data availability.* Polar vortex masks used in this manuscript are available at: https://nc.uni-bremen.de/index.php/s/yQFrsQwqPaRcdiD. Code to reproduce these vortex masks is available at: https://nc.uni-bremen.de/index.php/s/32f9SYNNRb8jeBD. The TCCON data were obtained from the TCCON Data Archive hosted by CaltechDATA at https://tccondata.org.



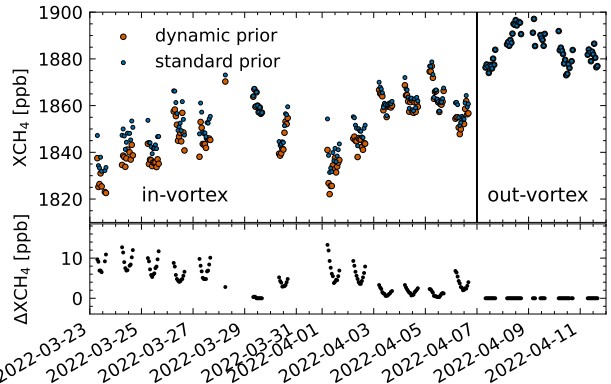

**Figure 22.** Comparison between the TCCON XCH$_4$ using the standard and dynamic prior for an exemplary time period at NYA. Data shown exhibits differences up to 17 ppb. The u-shapes in the $\Delta$XCH$_4$ reflect the airmass dependence present in the standard XCH$_4$ which has been reduced or removed in the dynamic prior XCH$_4$.

## Appendix A: Calculation of polar vortex mask

The polar vortex mask is calculated following the approach formulated by Nash et al. (1996) with some small modifications. As input data we use ECMWF ERA5 (Hersbach et al., 2017) on potential temperature levels. Specifically, we use the potential vorticity (PV) and the u-wind component.

The vortex boundary is calculated by analyzing the area enclosed by PV isolines. Specifically, a so-called equivalent latitude (EQL) $\Phi$ can be calculated for each PV isoline, which is defined as the latitude boundary for which the area of the spherical cap is equal to the are enclosed by the PV isoline:

$$\Phi_Q = \sin^{-1}\left(1 - \frac{A_Q}{2\pi R^2}\right) \tag{A1}$$

where $A_Q$ is the area with PV>$Q$ and $R$ is the radius of the earth. The PV $Q$ plotted against $\Phi$ has a distinct "S" shape in the vortex region. The maximum of the gradient $\frac{dQ}{d\Phi}$ can thus be used to identify the vortex boundary. Additionally, the vortex can be identified by the high wind speeds along its edge. Thus, the maximum of the average u-wind speed along the PV isolines is used as further identification of the vortex boundary.

The vortex edge can be defined by multiplying the PV gradient with the maximum average u-wind distribution, and choosing the maximum value in the vicinity of the PV maximum. The vortex boundary region is defined by the position of the local maximum and minimum of the second derivative of $Q$ below and above the vortex edge. For a more detailed explanation of these criteria, see Nash et al. (1996).

We perform our analysis on an evenly spaced grid of equivalent latitudes between 20–90°N, with a spacing identical to the input data (here 0.25°). To get the PV at each EQL value, Eq. A1 is numerically inversed. Specifically, we calculate the PV for which the area enclosed by the PV isoline is equal to the area corresponding to a given EQL. From these PV values, the



average u-wind is calculated as the mean u-wind between surrounding PV values. The distribution of $Q$, $\nabla Q$ and $\bar{u}$ can be seen in Fig. A1 as panels (a)–(c).

The vortex edge is now identified by multiplying the PV gradient distribution with the u-wind distribution (panel (d) in Fig. A1) and choosing the largest value in the vicinity of the PV gradient peak position ($\pm 5°$). A sloping filter is applied to the PV gradient values above $80°$, to suppress the high values in this region. Lastly, the vortex boundary region is defined by the

position of the minimum/maximum of the curvature of $Q$ (panel (e) in Fig. A1) above/below the position of the vortex edge within a vicinity of $\pm 5°$.

To identify whether the vortex is present on a given potential temperature level, we follow Nash et al. (1996) and utilize the maximum of the u-wind distribution (panel (c) in Fig A1), however we use a different threshold: the median of maximum u-wind speed.

Figure A2 shows the vortex edges and boundary regions derived from the above sketched algorithm. To determine whether a certain location is "inside" the polar vortex region, we combine the masks derived for all six potential temperature surfaces. Measurements which are outside the outer vortex boundary region in all masks are considered as "out-of-vortex". Measurements which are inside the vortex boundary on three or more masks are considered "inside-vortex". A python script to calculate the polar vortex edge from ERA5 data are available as a supplement to this manuscript.

## Appendix B: Static prior modification


Each $CH_4$ prior is modified by element-wise multiplication of the profile with a shifted normal distribution

$$\mathbf{f}(a,b,c,\mathbf{h}) = 1 + \frac{c}{b}\exp\left[-0.5\left(\frac{\mathbf{h}-a}{b}\right)^2\right] \tag{B1}$$

where $\mathbf{h}$ are the prior altitudes, $a$ is the mean of the distribution, $b$ the standard deviation or width of the distribution, and $c$ the magnitude of change in the prior profile. For the static prior, we use $a = 23$, $b = -4$ and $c = 1.5$. These values were determined

by testing various configurations for NYA.

## Appendix C: Dynamic prior modification

Each $CH_4$ prior is modified by element-wise multiplication of the profile with a shifted normal distribution

$$\mathbf{f}(a,b,c(c_{HF}),\mathbf{h}) = 1 + \frac{c}{b}\exp\left[-0.5\left(\frac{\mathbf{h}-a}{b}\right)^2\right] \tag{C1}$$

where $\mathbf{h}$ are the prior altitudes, $a$ is the mean of the distribution, $b$ the standard deviation or width of the distribution and $c$ the

magnitude of change in the prior profile. $c$ is dependent on the TCCON XHF value $c_{HF}$:

$$c(c_{HF}) = \begin{cases} c_{max} & , c_{HF} \geq 180 \\ \frac{c_{max}\cdot(c_{HF}-100)}{80} & , 180 > c_{HF} > 0 \\ 0 & , c_{HF} \leq 0 \end{cases} \tag{C2}$$



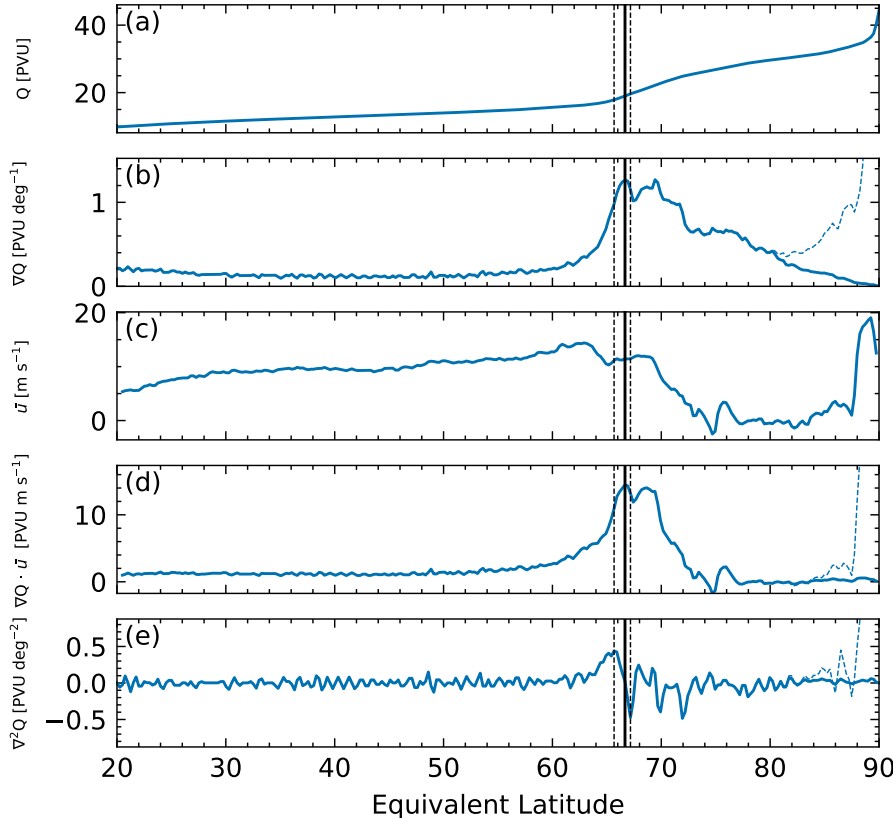

**Figure A1.** Overview of the values used to determine the polar vortex edge. These values were calculated for 22 April 2020 on the 430 K potential temperature level. Panel (a) shows the potential vorticity with the "S" shape visible around 70 EQL, (b) shows the gradient of the PV with a maximum around 70 EQL, (c) shows the u-wind component averaged along the PV isoline, (d) shows the product of the PV gradient and u-wind component and (e) shows the second derivative of the PV. The dashed blue lines show the distributions without the slope filter, which suppresses high PV values above 80 EQL. The black lines show the calculated vortex edge. The dashed black lines show the calculated boundary regions.

The form of Eq. C1 was informed by the difference between standard TCCON profiles and AirCores, NDACC and ACE-FTS profiles (see Figs. 6, 7 and 8). The form of Eq. C2 is based on the strong correlation between stratospheric $XCH_4$ and XHF. The minimum and maximum values were informed by looking at XHF values inside and outside the polar vortex. The other 490 parameters were set to $a = 28$, $b = -5$ and $c_{max} = 3$. These values were determined through testing and comparison to the stratospheric profile shape reported by AirCores or ACE-FTS.



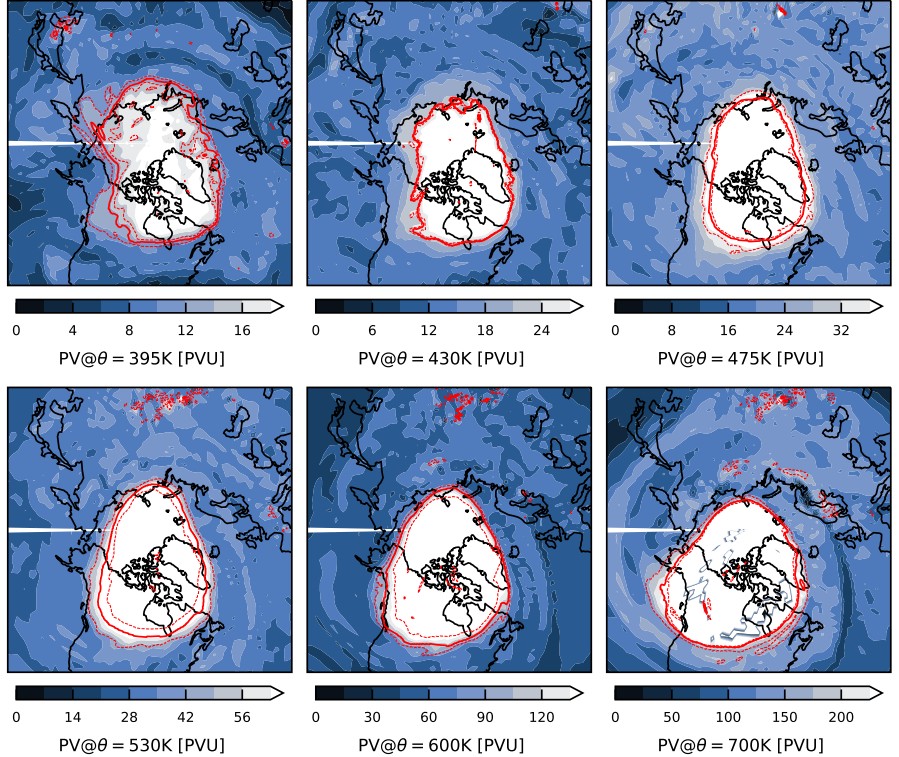

**Figure A2.** Polar vortex edges and boundary regions for the 19.03.2020. The red lines show the vortex edge, the dashed red lines the boundary regions. The contours in the background show the potential vorticity values on the corresponding potential temperature surfaces.

*Author contributions.* JH conceptualized this paper under supervision of MB with help from TW, JN and DW. The formal analysis was performed by JH. RK provided spectra for analysis. JH, DW and EM performed retrievals. JH wrote the initial draft of this manuscript. All authors provided constructive comments to improve the manuscript.

*Competing interests.* At least one of the (co-)authors is a member of the editorial board of Atmospheric Measurement Techniques.

*Disclaimer.* Contains modified Copernicus Climate Change Service information [2024]. Neither the European Commission nor ECMWF is responsible for any use that may be made of the Copernicus information or data it contains.

*Acknowledgements.* The Atmospheric Chemistry Experiment (ACE), also known as SCISAT, is a Canadian-led mission mainly supported by the Canadian Space Agency. The Eureka and Ny-Ålesund NDACC data used in this publication are available through the NDACC



website http://www.ndacc.org/. We gratefully acknowledge the provision of AirCore data from the Sodankylä polar vortex campaign in 2017 led by Rigel Kivi and Huilin Chen. We gratefully acknowledge the provision of TCOM-CH$_4$ data by Sandip Dhomse. We thankfully acknowledge the on-site support of AWIPEV personnel at AWIPEV research station in Ny-Ålesund, where the FTS measurements are performed under project number "AWIPEV_0004". The Eureka TCCON measurements were made at the Polar Environment Atmospheric Research Laboratory (PEARL) by the Canadian Network for the Detection of Atmospheric Change (CANDAC), primarily supported by the

Natural Sciences and Engineering Research Council of Canada (NSERC), Environment and Climate Change Canada, and the Canadian Space Agency. We thank the Canada Foundation for Innovation and NSERC for infrastructure and data analysis support for the TCCON station at East Trout Lake. RK acknowledges financial support by the Academy of Finland through grant number 140408 and by the European Space Agency (grant no. ESA-IPL-POE-LG-cl-LE-2015-1129).

*Financial support.*  This project is funded through the University of Bremen, as part of the junior research group "Greenhouse gases in the

Arctic". We gratefully acknowledge the funding by the Deutsche Forschungsgemeinschaft (DFG, German Research Foundation) – Projektnummer 268020496 – TRR 172, within the Transregional Collaborative Research Center "ArctiC Amplification: Climate Relevant Atmospheric and SurfaCe Processes, and Feedback Mechanisms (AC)³".



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
