# Peer review of "Reduction of airmass-dependent biases in TCCON XCH4 retrievals during polar vortex conditions"

_EGUsphere, 2024_

## Author Comment (AC1)

**Author's Response to Reviewer 1**

Jonas Hachmeister

July 14, 2025

We thank Frank Hase for reviewing the paper and providing valuable feedback.

**Reply to specific comments**

**I agree one would expect the variations of the CH4 a-priori profile to be a profound disturbance of CH4 retrievals under vortex conditions. However, the disappointingly low Pearson correlation of the results shown in Fig. 4 and the huge scatter of airmass dependence as fct of XGF shown in Figs. 10, 11, 12, 13, and 14 seem to indicate further mechanisms of action being involved. There is only a very short discussion on this problem (lines 145 ff). I think this aspect would deserve a more systematic investigation. Specifically, I would find it interesting to show the typical scatter of XCH4 airmass dependence for a midlatitude background site. This would provide a benchmark and help to decide whether this large scatter is related to some additional mechanism affecting polar sites.** We added a new appendix showing the airmass dependence as a function of XHF for two mid-latitude sites: Bremen and Orleans. This shows that (a) scatter in AMD is similar in the Arctic compared to mid-latitude sites and (b) that increased AMDs for larger XHF can also be observed at mid-latitude sites. (a) is expected as numerous effects can cause AMDs separate from problems with the prior profile. (b) indicates that prior related problems can also occur outside the Arctic.

**Overall, it would preferrable to perform a more consistent investigation across all TCCON sites (the model prior is investigated for Ny-Alesund only, why?).** The model prior was first tested for Ny-Ålesund and showed no improvement compared to the dynamic prior modification. Because the number of possible retrievals was limited by available resources and time, the model prior was not tested for the other sites. For Eureka, only the dynamic prior is provided due to technical problems. Modified retrievals were only possible after the manuscript was first submitted. This only allowed the inclusion of the dynamic prior modification during the first technical corrections.

**On several occurences (section on detection of polar vortex, use of model prior, relation between observation and vortex edge, ...) the reader wonders whether the slanted line-of-sight of the FTIR measurement is taken into account. Given the low SZA angles during relevant periods, the lateral displacement of LOS coordinates as function of altitude can be quite pronounced. Please detail on this aspect.** The TCCON retrieval assumes that the prior profiles derived for a given site is applicable to the slant column that the instrument really measures. If this assumption sufficiently violated this can lead to problems at all sites. In the Arctic, assuming a SZA of 75° the lateral displacement of the line of sight is about 37km/75km at a height of 10km/20km. Since the Arctic sites look southward this could lead to situations where the line of sight does not or only partially crosses the polar vortex even if the site is located within the vortex according to a mask. As of now this is not considered in the GINPUT software. Mention of this lateral displacement was added to the section "Polar vortex as based on Nash criterion" and "Detection of polar vortex air using a chemical tracer". Mention of the assumption that the prior is applicable to the slant column was added to the section "TCCON prior". Mention that the line of sight was not regarded was also added to the "Model prior" section. We added mention of the importance of the line of sight to the conclusions.

**My main critics of the current manuscript is related to section 7.4, the AirCore comparison. In my impression, the study falls short at this point. A single AirCore is used for illustrating the effects on a TCCON observation. I would expect a systematic investigation in this section which makes use of all available in-vortex AirCore launches and compares these profiles with standard TCCON a-prioris for estimating the expected disturbance on TCCON XCH4 results. Note that this only requires TCCON sensitivities, not actual colocated TCCON observations. Next, the static prior (using the option of a vortex mask) and the model prior could undergo the same kind of investigation.** In the manuscript we use AirCore data for Sodankylä between 2017–2021. Unfortunately, only two AirCores (both shown in the manuscript) with a clearly depleted stratospheric $CH_4$ profile are available from this dataset. The available measurement data of $CH_4$ profiles (known to us) is hence really sparse. We agree however that it is possible to estimate the expected effect on the TCCON retrieval by "simulating" a TCCON retrieval using a TCCON prior and averaging kernel and an arbitrary profile inside the vortex to perform a sensitivity study. However, we do not expect this to provide any additional insights compared to comparison of retrievals using different priors presented in the paper (see Fig. 22, which shows the magnitude of the XCH4 difference between different priors).

**Reply to minor/technical comments**

**Abstract: "In the Arctic .. polar nights .. prevent solar absorption measurements for half of the year". This is not true.** Correct, this was unclear as data coverage depends on the site latitude (and surrounding topography). This was changed to "...which prevent solar absorption measurements during large parts of the year".

**Abstract: "These effects can be explained by the fact that TCCON uses a profile scaling retrieval". This would indicate that application of a profile retrieval would altogether cure the problem. This is not true, as a constrained profile retrieval still has imperfect column sensitivity (although improved over a scaling retrieval).** That is correct, similar problems are expected for profile retrievals with imperfect vertical sensitivity. We changed this sentence accordingly: "These effects can be explained by the imperfect vertical sensitivity, especially to the stratosphere"

**Appendix B and C: Why are these rather ad-hoc profile correction schemes used? A correction describing a downwelling of the original undisturbed profile would better correspond to the underlying processes?** The intention of this paper was to identify, quantify and explain airmass-dependent biases in $XCH_4$ during polar vortex conditions. For this the ad-hoc profile correction scheme was sufficient to show that improvements to the $XCH_4$ retrieval are possible by modifying the prior. Ideally, we would have provided a complete solution to the problem (i.e., a new/updated prior generation scheme). This was not possible due to multiple reasons. First, the TCCON prior profiles are uniformly generated using the GINPUT software. Changes to the TCCON prior profiles can thus only be made by modifying the GINPUT software. This needs to be done in close collaboration with the TCCON. This manuscript could provide a starting point for such a discussion. Furthermore, it is not clear whether problems with the TCCON prior stem directly from the GINPUT software or from problems with the underlying used GEOS-FPIT model data.

---

## Author Comment (AC2)

**Author's Response to Reviewer 2**

Jonas Hachmeister

July 14, 2025

We thank Josh Laughner for reviewing the paper and providing valuable feedback.

**Reply to major comment**

**Please indicate whether the dynamic approach truly introduces a new prior for each spectrum or uses some average XHF value throughout the 3 hour block to modify the prior for a common set of spectra.[...]** This was not clearly communicated. We simply changed every prior for each 3 hour block as generated by gsetup. This is clarified in the updated manuscript.

**Second, I am very curious if the dynamic method must use column average XHF, or if the vertical column density of HF is sufficient.** We have performed a modified retrieval using the HF VCD instead of XHF for the Ny-Ålesund site. The results are shown in a new Appendix E and show similar results as the dynamic prior modification using XHF. Hence, we expect that the use of HF VCDs is sufficient.

**Reply to minor comments**

**Section 1: the introduction is a bit thin on why it is important to improve the retrievals for the relatively small number of arctic sites.[...]** We agree and added more motivation for our research to Section 1.

**Lines 78-80: "Trace gas measurements using remote sensing techniques based on solar absorption spectroscopy (like TCCON or various satellites) are expected to be affected by the polar vortex only in (early) spring, when sufficient light again becomes available to conduct measurements, as the vortex needs time to fully form during the autumn." This is true for sites above the arctic circle, but the fact that you include ETL in this study shows that there is also concern about vortex filaments reaching sites outside the arctic circle, and**

those sites would be affected throughout their winter season. Recommend making this statement more general to capture more of the relevant cases. We agree and made the statement more general.

Sect. 3.1.1: The TCCON retrievals use GEOS FP-IT or GEOS IT data. While those are not easily accessible, GEOS FP is, and that is a more similar product to the standard TCCON meteorological inputs. It includes Ertel's potential vorticity and wind variables, so a note on why you chose ERA5 data over GEOS FP would be helpful. (Perhaps because GEOS FP does not cover the full operational time span for Ny Alesund?) Again, from the perspective of making this operational, we would need to know whether there is a compelling reason to investigate ERA5 met data as an alternative for future algorithm versions. The polar vortex mask used in this manuscript is based on the Nash criterion which necessitates the use of potential vorticity values and u-wind on potential temperature layers. For this purpose ERA5 data was the easiest to access for us. Apart from that there is no reason that would favor ERA5 data compared to GEOS IT/FP-IT data. If these data are also available on potential temperature levels polar vortex mask calculations are similarly possible. Calculations of the polar vortex mask were performed on a standard computer with 8 CPU cores and 64GB RAM and do not need special resources.

Lines 120-122: "AMDs can be caused by uncertainties in spectroscopy, by instrument alignment, by non-linearity problems and by the use of the wrong measurement time. TCCON data are corrected during post-processing using an airmass-dependent correction factor..." To be specific, the airmass correction is intended to correct an airmass dependence that is consistent across all sites (which should come from errors in the spectroscopy). Issues of non-linearity and timing errors should be corrected by individual sites earlier in the retrieval process, and severely mis-aligned spectra should be flagged out. Please rephrase this to clarify that the airmass correction is targeted at the spectroscopically-driven airmass dependences only, and the other factors should be handled with their own correction procedures. We clarified this in the updated manuscript.

Line 132: "We define the AMD as the slope of the linear function fitted to the XCH4-SZA data within a day." Please indicate if you use the 82 deg maximum SZA limit typically applied to TCCON data. If not, it might be worth addressing why you use SZA instead of airmass as the predictor, since at very large SZAs, the relationship between the two becomes more non-linear, and airmass should have the more direct physical relationship to the deviation in XCH4. The analysis in Section 4 is based on publicly available TCCON data. Throughout the manuscript only data passing the quality filter (i.e., spectra that are included

in the public netcdf files) are used in the analysis. Hence, the SZA are limited to 82°.

**Line 143: "A clear tendency of higher AMD for higher XHF (and hence inside-vortex air) can be seen..." Perhaps qualify that this is clearest at the high latitude sites (NYA, EUR, SOD), with ETL being more ambiguous.** We adapted this sentence in the updated manuscript accordingly.

**Lines 147-150: "This can be explained by a) other effects causing AMD, which have not been corrected by the airmass-dependent correction factor and are not considered here, b) the existing prior not being consistently wrong (the difference between prior and true profile shape can vary) or c) true changes in diurnal XCH4 caused by local emissions or changes in atmospheric transport." (c) is why the procedure to derive the airmass corrections for the TCCON retrieval fit basis functions that are both symmetrical and asymmetrical with respect to solar noon. It is not perfect, but could address this issue. A note explaining why you did not use the standard TCCON fitting approach would be appropriate.** We fit a 1st degree polynomial to the SZA-XCH4 data as the simplest method to estimate AMD. This does not capture potential diurnal XCH4 variations. This is justified twofold: First, it keeps things simple (and avoids potential artificial reduction of AMD if using more complex functions). Second, data coverage in the high-latitudes is often limited. A linear fit of the (potentially) few data points is hence more stable than fitting a 2nd degree polynomial (or more complex functions) to the data. And lastly, this can also be justified by the relative isolation of the high-latitude sites which should keep diurnal XCH4 variations due to transport from source regions minimal (The exception in this regard could be ETL which is not as isolated as the other sites).

**Fig. 4 and 22: it is very difficult to distinguish the two series of points by size alone. Please consider using different marker types (e.g., + and o).** We updated both figures accordingly.

**Fig. 4: I assume "rho" in the legend is the coefficient represented by "R" in other literature, i.e., a value of 1 is perfect correlation and -1 is perfect anticorrelation? If so, please use "R" rather than "rho"; "rho" is too easily confused with "p" as in the p-statistic referenced in statements like "the slope is significant at the p = 0.05 confidence level".** Yes "rho" is the Pearson correlation coefficient. We changed "rho" to "R" as suggested.

**Lines 190-192: "To enable direct comparison between NDACC profiles and TCCON priors (see Sec. 5.4), the closest TCCON mea-**

surement within a day was collocated to each NDACC measurement." Please provide a scatter plot (in an SI or appendix would be fine) showing the NDACC vs. TCCON observation times that were matched. This would allow the reader to understand how close in time these values are if, e.g., a site does NDACC measurements in the morning and TCCON measurements in the afternoon. We create a scatter plot showing the different observations times. This will be added as a supplementary figure (S3).

Figs. 6 and 7: Please make the lines in the legend thicker; it is difficult to see the line colors in the legend clearly with such thin lines. Also recommend moving the legend outside of the figure and increasing the font size. Both figures were made more readable by updating them accordingly.

Sect. 6.3: Why was the model prior only tested for Ny-Alesund? It would be helpful to know if this model is an option for other arctic sites. The model prior was first tested for Ny-Ålesund and showed no improvement compared to the dynamic prior modification. For an operationalized implementation the model prior is also not feasible due to the new dependency on an external (and not necessarily regularly updated) data product. Because of this and because the number of possible retrievals was limited by available resources and time, the model prior was not tested for the other sites.

Lines 266-268: "Retrievals using modified priors were performed for NYA, SOD, ETL and EUR. Retrievals using the static priors were performed for NYA, SOD and ETL. Retrievals using the dynamic prior were performed for all three stations. The model prior was only tested for NYA." From results later in the paper, it looks like the dynamic prior was tested on Eureka data, but these three sentences make it sound like the dynamic prior was only tested on NYA, SOD, and ETL. It would also be worth mentioning why EUR did not test the static priors. Thank you for catching this mistake. The dynamic prior was indeed tested for all four stations. For Eureka, modified retrievals were only possible after the manuscript was first submitted. This only allowed the inclusion of the dynamic prior modification during the first technical corrections. The static prior was not tested due to time and resource constraints. Explanation why not all priors were tested for all sites was added to the text.

Lines 277-278: "The static prior was especially designed for inside-vortex measurements and thus yields a significant bias for high-XHF measurements…" Should "significant bias" be "significant bias reduction"? More generally, I suggest avoiding the use of "bias" here; that implies knowledge of the systematic difference between the retrieved and true XCH4. While the reduction in airmass dependence

is a good indicator that the retrievals will be more accurate, it is only an indirect metric. Perhaps instead you might say a "significant reduction in AMD" (and note the first time that this likely indicates a more accurate retrieval). We changed this part of the text as suggested.

Line 280: "...and leads to an overall improvement with values below $\mu = 1.06$ ppb deg$^{-1}$." Do you mean "leads to a lower mean AMD of $\mu = 1.06$ ppb deg$^{-1}$ for values with XHF<100 ppt"? Yes! We changed the sentences accordingly.

Lines 296-297: "Overall, the dynamic prior reduces the average AMD for most data for all four stations. For NYA, the dynamic prior shows the best results, while for SOD and ETL over corrections are visible for the range 140>XHF≥120 ppt." But this might be because you fit Ny-Alesund data to calculate the dynamic correction, yes? How much do the dynamic method's coefficients change if you fit data from the other stations? Does the station from which you derive the coefficients always have the best results? How might we think about ensuring the most representative correction for all arctic and subarctic sites if the coefficients vary too much depending on which sites' data are fit? Yes, the dynamic prior modification was initially developed for NYA data and was empirically derived by testing a range of different parameters. Ideally, derivation of these parameters is performed automatically from the data on a site-by-site basis. This was however out-of-scope for this manuscript, and here we wanted to test what improvements can be gained by applying a single correction to different sites. We would expect similar improvements (as for NYA) for the other sites if parameters are adapted. Potentially, further improvements are possible when using a more sophisticated method. We added mention that the better performance for NYA is expected to the text.

Sect 7.2: It would be helpful to include a figure, table, or discussion of whether the RMS/CL values for spectra that the XHF method classifies as in-vortex are actually out-vortex according to the EPV and wind mask, or vice versa (from the discussion around Fig. 2). This would be important to know, because if those false positives and negatives are the ones with the largest increase in RMS/CL, then that suggests that an operational implementation of this approach would benefit from including the vortex mask as a binary criterion on top of the XHF dependence modification. We tested this for NYA and only minor improvements (1-2%) are gained for the dynamic prior if using an additional vortex mask as a binary criterion. See supplementary figures S4 and S5. In Fig. S5 it can be seen that even out-of-vortex measurements improve using the dynamic prior which highlights the usefulness of the dynamic prior compared to a static modification in combination of a vortex mask.

**Lines 307-308, Figs. 15-18: "Positive values of $\Delta R$ constitute an improvement of the fit (lower RMS/CL), negative values an increase in RMS/CL compared to the reference retrieval." This seems backwards to me, (new - current)/current would be more intuitive so that negative values match up with a decrease in RMS/CL. Later, you use the (new - current)/current convention for the AKs, so being consistent would help the readers interpret the various plots more easily.** The different conventions are used to make each figure itself easier to understand. For example, in Fig. 20, a positive value corresponds to an increase in AK compared to the standard AK. And in Fig 15-18, a positive value corresponds to an improvement compared to the standard retrieval.

**Line 324-325: "Improvements are between roughly 53% and 72% for the different fit windows and thus smaller than for NYA." Meaning between 53% and 72% of the spectra have improved RMS/CL values? If so, please say that more explicitly.** Yes, we clarified this sentence.

**Lines 355-356: "where $\Delta Ai$ is the relative change of the AK. This yields differences up to 10 ppb in magnitude and a mean difference of roughly 3.5 ppb." It is worth putting this in the context of the TCCON error budget: since that is 4 to 4.5 ppb for XCH4, the mean is within our standard uncertainty. How common are the differences above the error budget? And what is the shape of the example profile used here?** The example profile is the extended AirCore profile shown in Fig. 21. Roughly 39% of the 1000 randomly sampled spectra exhibit values of $\Delta XCH_4$ larger than 4 ppb. We added mention of this and reference to the error budget to the text.

**Line 358: "Previous results were confined to the analysis of relative improvements between different versions of the TCCON retrieval." Recommend rephrasing, as this sounds like comparisons were done between major versions of the TCCON retrieval (e.g., GGG2014 vs. GGG2020) and possibly results in other papers. Perhaps instead: "The results in the previous sections were confined to the differences among retrievals using different a priori CH4 profiles."** We updated the sentence as suggested.

**Line 364: Was the AirCore integration done with a pressure weighting method? Please provide a reference or equation** Pressure weights were calculated using the formula provided by Connor et al. 2008. We added a reference to this paper to the text.

**Fig. 19 caption: "...the standard TCCON CH4 retrieval for NYA." Perhaps clearer to say "using the standard prior" to be consistent**

**with the language elsewhere in the paper.** We updated the caption as suggested.

**Lines 416-417: "Nonetheless, (i)–(iii) prove that improvements to the TCCON retrieval are possible using relatively simple modifications to the prior profile, which don't depend on external data." Please acknowledge that the dynamic method, in particular, adds a new back-dependency between the retrieved quantities and a priori profiles, which will require careful implementation to avoid poor quality HF retrievals from degrading the CH4 priors. That is, the method is conceptually simple, but does involve a more complex operational implementation.** We updated the text to include mention of the back-dependency, complexity and need for careful implementation.

**Figs. 19 & 20: these might be better combined into a single figure so that a reader can compare the standard AKs and the changes without having to switch pages.** We tried this before, however the figures become to small then. Hence, we leave the figures separated.

**Line 431: "In summary, we want to highlight that the prior shape has a significant impact on the retrieval..." Here again quantifying this relative to the TCCON error budget would be useful: changes on the order of twice the error budget are statistically significant and worth reducing, but do not mean that the current approach has a critical flaw.** We added mention of the error budget to the conclusions

**Fig. 22: is the difference dynamic minus standard or vice versa? Dynamic minus standard would follow the same (new - current) convention discussed previously and is my preference, and in either case, the sign convention should be stated.** Here (current-new) is used. We added mention of this to the figure caption.

**Code and data availability: Thank you for including a notebook to walk through the calculation of the vortex mask. I would also like to see at least the code used to derive and apply the static and dynamic modifications be included as well, so that it is archived in case we need to redo this analysis in the future for updated base CH4 profiles. It would also be good practice to include a requirements.txt, pyproject.toml, or environment.yml file alongside the code to identify the versions of Python packages used here.** Code for the generation of the static and dynamic prior modification will be added as additional supplementary code. We will also add a requirements.txt as suggested.